# Temporal Representation Alignment: Successor Features Enable Emergent Compositionality in Robot Instruction Following

**Vivek Myers**[*]    **Bill Chunyuan Zheng**[*]    **Anca Dragan**    **Kuan Fang**[1]    **Sergey Levine**

University of California, Berkeley    [1]Cornell University

## Abstract

Effective task representations should facilitate compositionality, such that after learning a variety of basic tasks, an agent can perform compound tasks consisting of multiple steps simply by composing the representations of the constituent steps together. While this is conceptually simple and appealing, it is not clear how to automatically learn representations that enable this sort of compositionality. We show that learning to associate the representations of current and future states with a temporal alignment loss can improve compositional generalization, even in the absence of any explicit subtask planning or reinforcement learning. We evaluate our approach across diverse robotic manipulation tasks as well as in simulation, showing substantial improvements for tasks specified with either language or goal images.

## 1   Introduction

Compositionality is a core aspect of intelligent behavior, describing the ability to sequence previously learned capabilities and solve new tasks [1]. In domains involving long-horizon decision-making like robotics, various learning approaches have been proposed to enable this property, including hierarchical learning [2], explicit subtask planning [3, 4, 5], and dynamic-programming-based "stitching" [6, 7]. In practice, these techniques are often unstable or data-inefficient in real-world robotics settings, making them difficult to scale [8].

By contrast, humans and animals are adept at quickly composing behaviors to reach new goals [1]. Possible explanations for these capabilities have been proposed, including the ability to perform transitive inference [9], learn successor representations and causal models [10, 11], and plan with world models [12]. In common among these theories is the idea of learning structured representations of the world, which inference about which actions will lead to certain goals.

How might these concepts translate to algorithms for robot learning? In this work, we study how adding an auxiliary successor representation learning objective affects compositional behavior in a real-world tabletop manipulation setting. We show that learning this representation structure improves the ability of the robot to perform long-horizon, compositionally-new tasks, specified either through goal images or natural language instructions. Perhaps surprisingly, we found that this temporal alignment does not need to be used for training the policy or test-time inference, as long as it is used as an auxiliary loss over the same representations used for the tasks (Fig. 1).

We compare our method, **T**emporal **R**epresentation **A**lignment (TRA), against past imitation and reinforcement learning baselines across a set of challenging multi-step manipulation tasks in the BridgeData setup [13] as well as the OGBench simulation benchmark [14]. Unlike prior work in setup, we focus on the compositional capabilities of the robot policies: as a whole, the tasks are

39th Conference on Neural Information Processing Systems (NeurIPS 2025).

Website: https://tra-paper.github.io/
[*]Equal contribution.

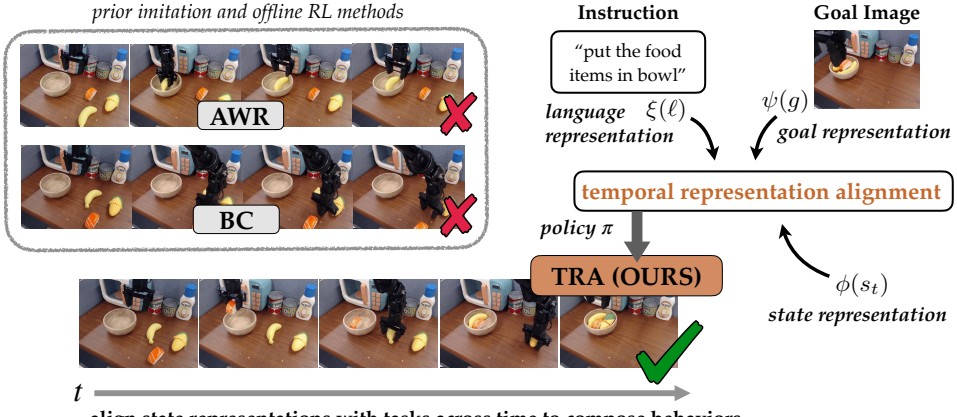

Figure 1: We show our Temporal Representation Alignment (TRA) method performing a language task, "put all food items in the bowl." TRA adds a time-contrastive loss for learning task representations to use with a goal- and language-conditioned policy. While TRA can implicitly decompose the task into steps and execute them one by one, the behavioral cloning (BC) and offline RL (AWR) methods fail at this compositional task. The structured representations learned by TRA enable this compositional behavior without explicit planning or hierarchical structure.

out-of-distribution, but each distinct subtask can be described through a goal image that lies in the training distribution. Adding a simple time-contrastive alignment loss improves compositional performance by >40% across 13 tasks in 4 evaluation scenes, and simulation results show better performance compared to behavioral cloning (i.e., no structured representation learning), and comparable performance to offline RL methods that explicitly use a learned value function.

## 2   Related Work

Our approach builds upon prior work on goal- and language-conditioned control, focusing particularly on the problem of compositional generalization.

**Robot manipulation with language and goals.**   Recent improvements in robot learning datasets have enabled the development of robot policies that can be commanded with image goals and language instructions [5, 13, 15]. These policies can be trained with goal- and language-conditioned imitation learning from human demonstrations [16, 17, 18, 19, 20], reinforcement learning [21, 22], or other forms of supervision [23, 24]. When trained to reach goals, methods can additionally use hindsight relabeling [25, 26] to improve performance [13, 27, 28, 29]. Our work shows how the benefits of goal-conditioned and language-conditioned supervised learning can be combined with temporal representation alignment to enable compositionality that would otherwise require planning or reinforcement learning.

**Compositional generalization in sequential decision making.**   In the context of decision making, compositional generalization refers to the ability to generalize to new behaviors that are composed of known sub-behaviors [30, 31]. Biological learning systems show strong compositional generalization abilities [9, 28, 32, 33], and recent work has explored how similar capabilities can be achieved in artificial systems [34, 35, 36]. In the context of policy learning, exploiting the compositionality of the behaviors can lead to generalization to unseen and temporally extended tasks [6, 37, 38, 4, 39, 40].

Hierarchical and planning-based approaches also aim to enable compositional behavior by explicitly partitioning a task into its components [41, 42, 43, 44]. With improvements in vision-language models (VLMs), many recent works have explored using a pre-trained VLM to decompose a task into subtasks that are more attainable for the low-level manipulation policy [5, 45, 46, 47, 42, 48, 49]. These approaches are limited by the need for robust pre-trained models that can be fine-tuned and prompted for embodied tasks. Our contribution is to show compositional properties can be achieved *without* any explicit hierarchical structure or planning, by learning a structured representation through time-contrastive representation alignment.

**Representation learning for states and tasks.** State and task representations for decision making aim to improve generalization and exploit additional sources of data. Recent work in the robotics domain have explored the use of pre-trained representations across multimodal data, including images and language, for downstream tasks [50, 51, 52, 27, 53, 54, 55, 56, 57]. In reinforcement learning problems, representations are often trained to predict future states, rewards, goals, or actions [58, 59, 60, 61], and can improve generalization and sample efficiency when used as value functions [62, 63, 64, 65, 66]. Some recent works have explored the use of additional structural constraints on representations to enable planning [41, 43, 67, 68, 69], or enforced metric properties to improve compositional generalization [70, 71, 72].

The key distinction between our approach and past contrastive representation methods for robotics like VIP [59], GRIF [27], and R3M [53] is that we focus on the real-world compositional generalization capabilities enabled by simply aligning representations across time in addition to the task modalities, without using the learned representations for policy extraction or defining a value function.

## 3    Temporal Representation Alignment

When training a series of short-horizon goal-reaching and instruction-following tasks, our goal is to learn a representation space such that our policy can generalize to a new (long-horizon) task that can be viewed as a sequence of known subtasks. We propose to structure this representation space by aligning the representations of states, goals, and language in a way that is more amenable to compositional generalization.

**Notation.** We take the setting of a goal- and language-conditioned MDP $\mathcal{M}$ with state space $\mathcal{S}$, continuous action space $\mathcal{A} \subseteq (0, 1)^{d_{\mathcal{A}}}$, initial state distribution $p_0$, dynamics $\mathrm{P}(s' \mid s, a)$, discount factor $\gamma$, and language task distribution $p_\ell$. A policy $\pi(a \mid s)$ maps states to a distribution over actions. We inductively define the $k$-step (action-conditioned) policy visitation distribution as:

$$p_1^\pi(s_1 \mid s_1, a_1) \triangleq p(s_1 \mid s_1, a_1),$$

$$p_{k+1}^\pi(s_{k+1} \mid s_1, a_1) \triangleq \int_{\mathcal{A}} \int_{\mathcal{S}} p(s_{k+1} \mid s, a) \, \mathrm{d}p_k^\pi(s \mid s_1, a_1) \, \mathrm{d}\pi(a \mid s)$$

$$p_{k+t}^\pi(s_{k+t} \mid s_t, a_t) \triangleq p^\pi(s_k \mid s_1, a_1). \tag{1}$$

Then, the discounted state visitation distribution can be defined as the distribution over $s^+$, the state reached after $K \sim \mathrm{Geom}(1 - \gamma)$ steps:

$$p_\gamma^\pi(s^+ \mid s, a) \triangleq \sum_{k=0}^{\infty} \gamma^k p_k^\pi(s^+ \mid s, a). \tag{2}$$

We assume access to a dataset of expert demonstrations $\mathcal{D} = \{\tau_i, \ell_i\}_{i=1}^K$, where each trajectory

$$\tau_i = \{s_{t,i}, a_{t,i}\}_{t=1}^H \in \mathcal{S} \times \mathcal{A} \tag{3}$$

is gathered by an expert policy $\pi^{\mathrm{E}}$, and is then annotated with $p_\ell(\ell_i \mid s_{1,i}, s_{H,i})$. Our aim is to learn a policy $\pi$ that can select actions conditioned on a new language instruction $\ell$. As in prior work [13], we handle the continuous action space by representing both our policy and the expert policy as an isotropic Gaussian with fixed variance; we will equivalently write $\pi(a \mid s, \varphi)$ or denote the mode as $\hat{a} = \pi(s, \varphi)$ for a task $\varphi$.

### 3.1    Representations for Reaching Distant Goals

We learn a goal-conditioned policy $\pi(a \mid s, g)$ that selects actions to reach a goal $g$ from expert demonstrations with behavioral cloning. Suppose we directly selected actions to imitate the expert on two trajectories in $\mathcal{D}$:

$$\left. \begin{array}{c} s_1 \longrightarrow s_2 \longrightarrow \ldots \longrightarrow s_H \longrightarrow w \\[1em] w \longrightarrow s_1' \longrightarrow \ldots \longrightarrow s_H' \longrightarrow g \end{array} \right\} \quad \tau_i \in \mathcal{D} \tag{4}$$

When conditioned with the composed goal $g$, we would be unable to imitate effectively as the composed state-goal $(s, g)$ is jointly out of the training distribution.

What *would* work for reaching $g$ is to first condition the policy on the intermediate waypoint $w$, then upon reaching $w$, condition on the goal $g$, as the state-goal pairs $(s_i, w)$, $(w, g)$, and $(s'_i, g)$ are all in the training distribution. If we condition the policy on some intermediate waypoint distribution $p(w)$ (or sufficient statistics thereof) that captures all of these cases, we can stitch together the expert behaviors to reach the goal $g$.

Our approach is to learn a representation space that captures this ability, so that a GCBC objective used in this space can effectively imitate the expert on the composed task. We begin with the goal-conditioned behavioral cloning [26] loss $\mathcal{L}_{\mathrm{BC}}^{\phi, \psi, \xi}$ conditioned with waypoints $w$.

$$\mathcal{L}_{\mathrm{BC}}\left(\{s_i, a_i, s_i^+, g_i\}_{i=1}^K\right) = \sum_{i=1}^K \log \pi\left(a_i \mid s_i, \psi(g_i)\right). \tag{5}$$

Enforcing the invariance needed to stitch Eq. (4) then reduces to aligning $\psi(g) \leftrightarrow \psi(w)$. The temporal alignment objective $\phi(s) \leftrightarrow \phi(s^+)$ accomplishes this indirectly by aligning both $\psi(w)$ and $\psi(g)$ to the shared waypoint representation $\phi(w)$:

$$\mathcal{L}_{\mathrm{NCE}}\left(\{s_i, s_i^+\}_{i=1}^K; \phi, \psi\right) = \log\left(\frac{e^{\phi(s_i^+)^T \psi(s_i)}}{\sum_{j=1}^K e^{\phi(s_i^+)^T \psi(s_j)}}\right) + \sum_{j=1}^K \log\left(\frac{e^{\phi(s_i^+)^T \psi(s_i)}}{\sum_{i=1}^K e^{\phi(s_i^+)^T \psi(s_j)}}\right) \tag{6}$$

## 3.2 Interfacing with Language Instructions

To extend the representations from Section 3.1 to compositional instruction following with language tasks, we need some way to ground language into the $\psi$ (future state) representation space. We use a similar approach to GRIF [27], which uses an additional CLIP-style [73] contrastive alignment loss with an additional pretrained language encoder $\xi$:

$$\mathcal{L}_{\mathrm{NCE}}\left(\{g_i, \ell_i\}_{i=1}^K; \psi, \xi\right) = \sum_{i=1}^K \log\left(\frac{e^{\psi(g_i)^T \xi(\ell_i)}}{\sum_{j=1}^K e^{\psi(g_i)^T \xi(\ell_j)}}\right) + \sum_{j=1}^K \log\left(\frac{e^{\psi(g_i)^T \xi(\ell_i)}}{\sum_{i=1}^K e^{\psi(g_i)^T \xi(\ell_j)}}\right) \tag{7}$$

## 3.3 Temporal Alignment

Putting together the objectives from Sections 3.1 and 3.2 yields the Temporal Representation Alignment (TRA) approach. TRA structures the representation space of goals and language instructions to better enable compositional generalization. We learn encoders $\phi, \psi$, and $\xi$ to map states, goals, and language instructions to a shared representation space.

$$\mathcal{L}_{\mathrm{NCE}}(\{x_i, y_i\}_{i=1}^K; f, h) = \sum_{i=1}^K \log\left(\frac{e^{f(y_i)^T h(x_i)}}{\sum_{j=1}^K e^{f(y_i)^T h(x_j)}}\right) + \sum_{j=1}^K \log\left(\frac{e^{f(y_i)^T h(x_i)}}{\sum_{i=1}^K e^{f(y_i)^T h(x_j)}}\right) \tag{8}$$

$$\mathcal{L}_{\mathrm{BC}}\left(\{s_i, a_i, s_i^+, \ell_i\}_{i=1}^K; \pi, \psi, \xi\right) = \sum_{i=1}^K \log \pi\left(a_i \mid s_i, \xi(\ell_i)\right) + \log \pi\left(a_i \mid s_i, \psi(s_i^+)\right) \tag{9}$$

$$\mathcal{L}_{\mathrm{TRA}}\left(\{s_i, a_i, s_i^+, g_i, \ell_i\}_{i=1}^K; \pi, \phi, \psi, \xi\right) \tag{10}$$

$$= \underbrace{\mathcal{L}_{\mathrm{BC}}\left(\{s_i, a_i, s_i^+, \ell_i\}_{i=1}^K; \pi, \psi, \xi\right)}_{\text{behavioral cloning}} + \underbrace{\mathcal{L}_{\mathrm{NCE}}\left(\{s_i, s_i^+\}_{i=1}^K; \phi, \psi\right)}_{\text{temporal alignment}} + \underbrace{\mathcal{L}_{\mathrm{NCE}}\left(\{g_i, \ell_i\}_{i=1}^K; \psi, \xi\right)}_{\text{task alignment}}$$

Note that the NCE alignment loss uses a CLIP-style symmetric contrastive objective [73, 67] — we highlight the indices in the NCE alignment loss (8) for clarity.

Our overall objective is to minimize Eq. (10) across states, actions, future states, goals, and language tasks within the training data:

$$\min_{\pi, \phi, \psi, \xi} \mathbb{E}_{\substack{(s_{1,i}, a_{1,i}, \ldots, s_{H,i}, a_{H,i}, \ell) \sim \mathcal{D} \\ i \sim \mathrm{Unif}(1\ldots H), k \sim \mathrm{Geom}(1-\gamma)}} \left[\mathcal{L}_{\mathrm{TRA}}\left(\{s_{t,i}, a_{t,i}, s_{\min(t+k,H),i}, s_{H,i}, \ell\}_{i=1}^K; \pi, \phi, \psi, \xi\right)\right]. \tag{11}$$

---

**Algorithm 1:** Temporal Representation Alignment

1: **input:** dataset $\mathcal{D} = (\{s_{t,i}, a_{t,i}\}_{t=1}^{H}, \ell_i)_{i=1}^{N}$
2: initialize networks $\Theta \triangleq (\pi, \phi, \psi, \xi)$
3: **while** training **do**
4:      sample batch $\left\{(s_{t,i}, a_{t,i}, s_{t+k,i}, \ell_i)\right\}_{i=1}^{K} \sim \mathcal{D}$
        for $k \sim \text{Geom}(1-\gamma)$
5:      $\Theta \leftarrow \Theta - \alpha \nabla_{\Theta} \mathcal{L}_{\text{TRA}}\left(\{s_{t,i}, a_{t,i}, s_{t+k,i}, \ell_i\}_{i=1}^{K}; \Theta\right)$
6: **output:** language-conditioned policy $\pi(a_t|s_t, \xi(\ell))$, goal-conditioned policy $\pi(a_t|s_t, \psi(g))$

---

### 3.4 Implementation

A summary of our approach is shown in Algorithm 1. In essence, TRA learns three encoders: $\phi$, which encodes states, $\psi$ which encodes future goals, and $\xi$ which encodes language instructions. Contrastive losses are used to align state representations $\phi(s_t)$ with future goal representations $\psi(s_{t+k})$, which are in turn aligned with equivalent language task specifications $\xi(\ell)$ when available. We then learn a behavior cloning policy $\pi$ that can be conditioned on either the goal or language instruction through the representation $\psi(g)$ or $\xi(\ell)$, respectively.

### 3.5 Temporal Alignment and Compositionality

We will formalize the intuition from Section 3.1 that TRA enables compositional generalization by considering the error on a "compositional" version of $\mathcal{D}$, denoted $\mathcal{D}^*$. Using the notation from Eq. (3), we can say $\mathcal{D}$ is distributed according to:

$$\mathcal{D} \triangleq \mathcal{D}^H \sim \prod_{i=1}^{K} p_0(s_{1,i}) p_\ell(\ell_i \mid s_{1,i}, s_{H,i}) \prod_{t=1}^{H} \pi^{\text{E}}(a_{t,i} \mid s_{t,i}) \, \text{P}(s_{t+1,i} \mid s_{t,i}, a_{t,i}), \quad (12)$$

or equivalently

$$\mathcal{D}^H \sim \prod_{i=1}^{K} p_0(s_{1,i}) p_\ell(\ell_i \mid s_{1,i}, s_{H,i}) \prod_{t=1}^{H} e^{\sigma^2 \|\pi^{\text{E}}(s_{t,i}) - a_{t,i}\|^2} \, \text{P}(s_{t+1,i} \mid s_{t,i}, a_{t,i}), \quad (13)$$

by the isotropic Gaussian assumption. We will define $\mathcal{D}^* \triangleq \mathcal{D}^{H'}$ to be a longer-horizon version of $\mathcal{D}$ extending the behaviors gathered under $\pi^{\text{E}}$ across a horizon $\alpha H \geq H' \geq H$ that additionally satisfies a "time-isotropy" property: the marginal distribution of the states is uniform across the horizon, i.e., $p_0(s_{1,i}) = p_0(s_{t,i})$ for all $t \in \{1 \dots H'\}$.

We will relate the in-distribution imitation error $\text{ERR}(\bullet; \mathcal{D})$ to the compositional out-of-distribution imitation error $\text{ERR}(\bullet; \mathcal{D}^*)$. We define

$$\text{ERR}(\hat{\pi}; \tilde{\mathcal{D}}) = \mathbb{E}_{\tilde{\mathcal{D}}}\Big[\frac{1}{H} \sum_{t=1}^{H} \mathbb{E}_{\hat{\pi}} \left[\|\tilde{a}_{t,i} - \hat{\pi}(\tilde{s}_{t,i}, \tilde{s}_{H,i})\|^2 / d_{\mathcal{A}}\right]\Big] \quad \text{for} \quad \{\tilde{s}_{t,i}, \tilde{a}_{t,i}, \tilde{\ell}_i\}_{t=1}^{H} \sim \tilde{\mathcal{D}}. \quad (14)$$

On the training dataset this is equivalent to the expected behavioral cloning loss from Eq. (9).

**Assumption 1.** *The policy factorizes through inferred waypoints as:*

$$\text{goals: } \pi(a \mid s, g) = \int \pi(a \mid s, w) \, \text{P}(s_t = w \mid s_{t+k} = g) \, \mathrm{d}w \quad (15)$$

$$\text{language: } \pi(a \mid s, \ell) = \int \pi(a \mid s, w) \, \text{P}(s_t = w \mid s_{t+k} = g) \, \text{P}(s_{t+k} = g \mid \ell) \, \mathrm{d}w \, \mathrm{d}g, \quad (16)$$

*where denote by $\pi(s, g)$ the MLE estimate of the action $a$.*

**Theorem 1.** *Suppose $\mathcal{D}$ is distributed according to Eq. (12) and $\mathcal{D}^*$ is distributed according to Eq. (12). When $\gamma > 1 - 1/H$ and $\alpha > 1$, for optimal features $\phi$ and $\psi$ under Eq. (11), we have*

$$\text{ERR}(\pi; \mathcal{D}^*) \leq \text{ERR}(\pi; \mathcal{D}) + \frac{\alpha - 1}{2\alpha} + \left(\frac{\alpha - 2}{2\alpha}\right) \mathbb{1}\{\alpha > 2\}. \quad (17)$$

We can also define a notion of the language-conditioned compositional generalization error:

$$\mathrm{ERR}^{\ell}(\pi; \mathcal{D}^*) \triangleq \mathbb{E}_{\mathcal{D}^*}\left[\frac{1}{H}\sum_{t=1}^{H}\mathbb{E}_{\pi}\left[\|\tilde{a}_{t,i} - \pi(\tilde{s}_{t,i}, \tilde{\ell}_i)\|^2\right]\right].$$

**Corollary 1.1.** *Under the same conditions as Theorem 1,*

$$\mathrm{ERR}^{\ell}(\pi; \mathcal{D}^*) \leq \mathrm{ERR}^{\ell}(\pi; \mathcal{D}) + \frac{\alpha - 1}{2\alpha} + \left(\frac{\alpha - 2}{2\alpha}\right)\mathbb{1}\{\alpha > 2\}.$$

The proofs as well as a visualization of the bound are in Appendix F. Policy implementation details can be found in Appendix B

## 4 Experiments

We ask the following questions:

1. Can TRA enable zero-shot composition of tasks without additional rewards or planning?
2. Does TRA improve compositional generalization over past methods?
3. How well does TRA capture skills that are less common within the dataset?
4. Is temporal alignment by itself sufficient for effective compositional generalization?

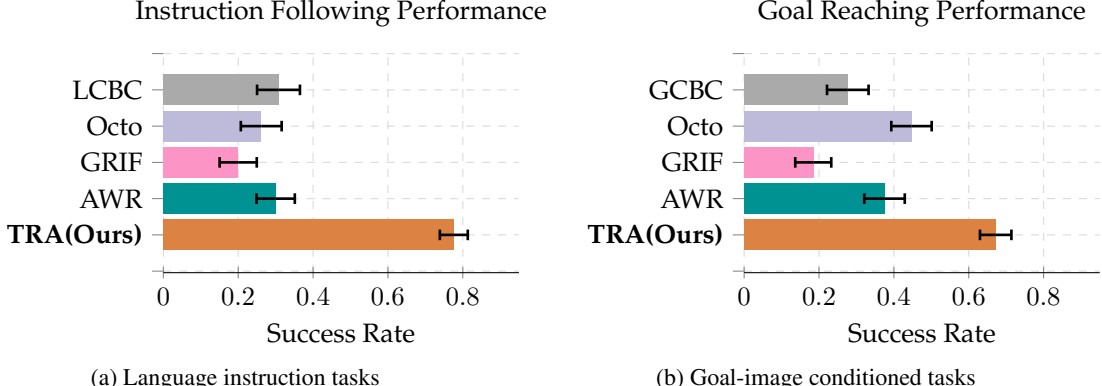

Figure 2: Aggregated performance on compositional generalization tasks, consisting of instruction-following and goal-reaching tasks.

### 4.1 Real-World Experimental Setup

We evaluate TRA on a collection of held-out *compositionally-OOD* tasks — tasks for which the individual substeps are represented in the dataset, but the combination of those steps is unseen. For example, in a task such as "removing a bell pepper from a towel, and then sweep the towel", both the tasks "remove the bell pepper from the towel" and "sweep the towel" have similar entries within BridgeData, but such a combination of behaviors is unseen. We utilize a real-world robot manipulation interface with a 7 DoF WidowX250 manipulator arm with 5Hz execution frequency. We train on an augmented version of the BridgeDataV2 dataset [13], which contains over 50k trajectories with 72k language annotations. More details are in Appendix B.

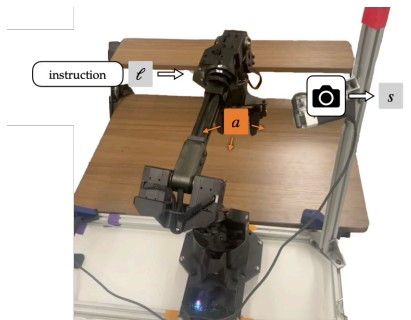

Figure 3: The tabletop manipulation setup used for the real-world evaluation (see 13).

In order to specifically test the ability of TRA to perform compositional generalization, we organize our evaluation tasks into 4 scenes that are unseen in BridgeData, each with increasing difficulty:

Set (A) – **One-Step:** These are the only tasks that are not compositionally-OOD, as all the tasks are one-step tasks. These tasks involve opening, putting an item in, and closing a drawer, and have been seen in BridgeData, although at a lower frequency than object manipulation, and with new positions.

Set (B) – **Task Concatenation:** These tasks scene involves concatenating multiple tasks of the same nature in sequence, where a robot must be able to perform all tasks within the same trajectory. During evaluation, we instruct the policy with instructions such as sweeping multiple objects in the scene that require composition (though are not sensitive to the *order* of the composition).

Set (C) – **Semantic Generalization:** Unlike set (B), these tasks require manipulation with different objects of the same type. We test this using various food items within BridgeData, instructing the policy within a container.

Set (D) – **Tasks with Dependency:** This is the most challenging set: these tasks have subtasks that require previous subtasks to be completed, such as taking an object out of a drawer.

The complete list of tasks is described in Appendix D.

## 4.2 Baselines

We compare against the following baselines in our real-world evaluation: **GRIF** [27] learns a goal- and language- conditioned policy using aligned goal image and language representations. In our experiments, this becomes equivalent to TRA when the temporal alignment objective is removed. **GCBC** [13] learns a goal-conditioned behavioral cloning policy that concatenates the goal image with the image observation. **LCBC** [13] learns a language-conditioned policy that concatenates the language with the image observation. **OCTO** [74] uses a multimodal transformer to learn a goal- and language-conditioned policy. The policy is trained on the Open-X dataset [75], which incorporates BridgeData in its entirety. **AWR** [76] uses advantages produced by a value function to effectively extract a policy from an offline dataset. In our experiments, we use the difference between the contrastive loss between the current observation and the goal representation and the contrastive loss between the next observation and the goal representation as a surrogate for value function.

Table 1: Real-world Evaluation

| | Task | Language-conditioned | | | | | Goal-conditioned | | | | |
|---|---|---|---|---|---|---|---|---|---|---|---|
| | | TRA | GRIF | LCBC | Octo | AWR | TRA | GRIF | GCBC | Octo | AWR |
| **(A)** | open the drawer | $0.80^{(\pm0.1)}$ | $0.20^{(\pm0.2)}$ | $0.60^{(\pm0.2)}$ | $0.60^{(\pm0.2)}$ | $0.40^{(\pm0.2)}$ | $0.60^{(\pm0.2)}$ | $0.60^{(\pm0.2)}$ | $0.40^{(\pm0.2)}$ | $0.50^{(\pm0.2)}$ | $0.80^{(\pm0.2)}$ |
| **(A)** | mushroom in drawer | $0.80^{(\pm0.1)}$ | $0.80^{(\pm0.2)}$ | $0.40^{(\pm0.2)}$ | $0.00^{(\pm0.0)}$ | $0.60^{(\pm0.2)}$ | $0.90^{(\pm0.1)}$ | $0.40^{(\pm0.2)}$ | $0.80^{(\pm0.2)}$ | $0.90^{(\pm0.1)}$ | $0.60^{(\pm0.2)}$ |
| **(A)** | close drawer | $0.60^{(\pm0.2)}$ | $0.60^{(\pm0.2)}$ | $0.40^{(\pm0.2)}$ | $0.60^{(\pm0.2)}$ | $0.40^{(\pm0.2)}$ | $1.00^{(\pm0.0)}$ | $0.40^{(\pm0.2)}$ | $0.80^{(\pm0.2)}$ | $0.60^{(\pm0.2)}$ | $0.40^{(\pm0.2)}$ |
| **(D)** | take the item out of the drawer | $0.60^{(\pm0.2)}$ | $0.00^{(\pm0.0)}$ | $0.00^{(\pm0.0)}$ | $0.20^{(\pm0.2)}$ | $0.00^{(\pm0.0)}$ | $0.40^{(\pm0.2)}$ | $0.00^{(\pm0.0)}$ | $0.00^{(\pm0.0)}$ | $0.20^{(\pm0.2)}$ | $0.00^{(\pm0.0)}$ |
| **(B)** | put the spoons on towels | $0.80^{(\pm0.1)}$ | $0.40^{(\pm0.2)}$ | $0.20^{(\pm0.2)}$ | $0.00^{(\pm0.0)}$ | $0.20^{(\pm0.2)}$ | $1.00^{(\pm0.0)}$ | $0.20^{(\pm0.2)}$ | $0.60^{(\pm0.2)}$ | $0.40^{(\pm0.2)}$ | $0.60^{(\pm0.2)}$ |
| **(B)** | put the spoons on the plates | $0.90^{(\pm0.1)}$ | $0.20^{(\pm0.2)}$ | $0.20^{(\pm0.2)}$ | $0.20^{(\pm0.2)}$ | $0.00^{(\pm0.0)}$ | $0.90^{(\pm0.1)}$ | $0.00^{(\pm0.0)}$ | $0.40^{(\pm0.2)}$ | $0.00^{(\pm0.0)}$ | $0.80^{(\pm0.2)}$ |
| **(C)** | put the corn and sushi on plate | $0.90^{(\pm0.1)}$ | $0.00^{(\pm0.0)}$ | $0.40^{(\pm0.2)}$ | $0.00^{(\pm0.0)}$ | $0.50^{(\pm0.2)}$ | $0.70^{(\pm0.1)}$ | $0.00^{(\pm0.0)}$ | $0.20^{(\pm0.2)}$ | $0.00^{(\pm0.0)}$ | $0.30^{(\pm0.1)}$ |
| **(C)** | sushi and mushroom in bowl | $0.80^{(\pm0.1)}$ | $0.00^{(\pm0.0)}$ | $0.60^{(\pm0.2)}$ | $0.20^{(\pm0.2)}$ | $0.60^{(\pm0.2)}$ | $0.50^{(\pm0.2)}$ | $0.00^{(\pm0.0)}$ | $0.20^{(\pm0.2)}$ | $0.40^{(\pm0.2)}$ | $0.60^{(\pm0.2)}$ |
| **(C)** | corn, banana, and sushi in bowl | $0.80^{(\pm0.1)}$ | $0.00^{(\pm0.0)}$ | $0.00^{(\pm0.0)}$ | $0.00^{(\pm0.0)}$ | $0.20^{(\pm0.1)}$ | $0.50^{(\pm0.2)}$ | $0.00^{(\pm0.0)}$ | $0.00^{(\pm0.0)}$ | $0.40^{(\pm0.2)}$ | $0.50^{(\pm0.2)}$ |
| **(D)** | corn on plate then sushi in pot | $0.70^{(\pm0.1)}$ | $0.00^{(\pm0.0)}$ | $0.40^{(\pm0.2)}$ | $0.60^{(\pm0.2)}$ | $0.20^{(\pm0.2)}$ | $0.30^{(\pm0.1)}$ | $0.20^{(\pm0.2)}$ | $0.00^{(\pm0.0)}$ | $0.00^{(\pm0.0)}$ | $0.00^{(\pm0.0)}$ |
| **(A)** | sweep to the right | $0.80^{(\pm0.1)}$ | $0.20^{(\pm0.2)}$ | $0.40^{(\pm0.2)}$ | $0.40^{(\pm0.2)}$ | $0.00^{(\pm0.0)}$ | $0.70^{(\pm0.1)}$ | $0.40^{(\pm0.2)}$ | $0.00^{(\pm0.0)}$ | $0.80^{(\pm0.2)}$ | $0.00^{(\pm0.0)}$ |
| **(B)** | fold cloth into the center | $1.00^{(\pm0.0)}$ | $0.20^{(\pm0.2)}$ | $0.40^{(\pm0.2)}$ | $0.40^{(\pm0.2)}$ | $0.40^{(\pm0.2)}$ | $0.80^{(\pm0.1)}$ | $0.00^{(\pm0.0)}$ | $0.00^{(\pm0.0)}$ | $0.60^{(\pm0.2)}$ | $0.00^{(\pm0.0)}$ |
| **(B)** | move bell pepper and sweep towel | $0.50^{(\pm0.2)}$ | $0.00^{(\pm0.0)}$ | $0.00^{(\pm0.0)}$ | $0.20^{(\pm0.2)}$ | $0.00^{(\pm0.0)}$ | $0.60^{(\pm0.2)}$ | $0.20^{(\pm0.2)}$ | $0.20^{(\pm0.2)}$ | $0.40^{(\pm0.2)}$ | $0.00^{(\pm0.0)}$ |

(A) One step tasks    (B) Task concatenation    †The best-performing method(s) up to statistical significance are **highlighted**
(C) Semantic generalization    (D) Tasks with dependency

We train GRIF, GCBC, LCBC, and AWR using the same augmented Bridge Dataset as TRA, and we use an Octo-Base 1.5 model for our evaluation. A more detail approach is detailed in Appendix C. During evaluation, we give all policies the same goal state and language instruction regardless of the architecture, as they are trained on the same language instruction with the exception of Octo, which doesn't benefit from paraphrased language data, but does benefit from a more diverse language annotation set across a larger dataset of varying length and complexity.

## 4.3 Real-world Evaluation

Our real-world evaluation aims to answer the following questions.

**Does TRA enable compositionality?** 1 shows the success rates of the TRA method compared to other methods on real-world robot evaluation tasks. We marked all policies within the task orange if they achieve the best statistically significant performance, as determined by a one-sided t-test with a significance level of 0.05 (see Appendix H for details). We first compare the performance against methods in (A). Although TRA performs well with drawer tasks, its performance against baseline methods is not statistically significant. However, TRA performs considerably better than that of any baseline methods on compositionally-OOD **instruction following** tasks.

While TRA completed 88.9% of tasks seen in (B), 83.3% of evaluations in (C), and 60% of tasks in (D) with instruction following, the best-performing baseline for (B) was 30% with LCBC, 43.3% for (C) with AWR, and 33.3% on (D) with Octo. The same improvement was also present in goal reaching tasks, although at a lower level, in which (C) produced 60% success rate and scene D produced a 43.3% success rate, as compared to 46.7% and 20% for the best baselines.

**How does TRA compare to conventional offline RL?** While offline reinforcement conventionally is considered necessary for "stitching" [77], we demonstrate that TRA still outperforms offline reinforcement learning on robotic manipulation. TRA performs better than AWR for both language and image tasks, outperforming AWR by 45% on instruction following tasks, and by 25% on goal reaching tasks, showing considerable improvement over an offline RL method that promises compositional generalization via stitching.

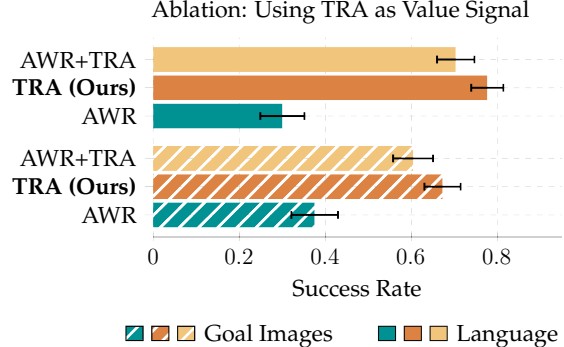

Figure 4: Aggregated success rate of using AWR as an additional policy learning metric over all 4 scenes.

The policy trained with AWR often stops after one subtask, even though the goal instruction or image demanded all of the subtasks be completed. We see this in, e.g., Fig. 1, where 3 different policies use the same goal image for a task where all 3 food items must be put in the bowl.

**Does TRA help rarely-seen skills within the dataset?** We also compare TRA against AWR across challenging tasks in the dataset. When conditioning on language, AWR struggles to to effectively generalize to compositionally harder tasks, with average success rate decreasing from 43.3% in to 6.67% from (C) to (D), compared to a decrease of only 83.3% to 60% for TRA. Other agents do not perform as well as AWR in (D), as the lack of such compositional generalization prevented the policies from achieving all of the tasks at a reliable rate.

**Is TRA sufficient in achieving compositional generalization?** We demonstrate in our real-world experiment that only using temporal alignment is sufficient for achieving good compositional generalization. We evaluate this by comparing a policy trained on only temporal alignment loss (our method), and another policy trained on such loss and have these losses weighed by AWR. Our AWR implementation is detailed in Appendix C.

Figure 4 shows that across all evaluation tasks, AWR provides no additional benefit on top of temporal alignment. In fact, using AWR marginally decreases the efficacy of TRA, unlike when used with GCBC and LCBC.

Table 2: OGBench Evaluation

| Task | TRA | GCBC | CRL | GCIQL | GCIVL | QRL |
|---|---|---|---|---|---|---|
| | | | Methods | | | |
| antmaze medium stitch | $60.7^{(\pm 3.0)*}$ | $45.5^{(\pm 3.9)}$ | $52.7^{(\pm 2.2)}$ | $29.3^{(\pm 2.2)}$ | $44.1^{(\pm 2.0)}$ | $59.1^{(\pm 2.4)}$ |
| antmaze large stitch | $12.8^{(\pm 2.0)}$ | $3.4^{(\pm 1.0)}$ | $10.8^{(\pm 0.6)}$ | $7.5^{(\pm 0.7)}$ | $18.5^{(\pm 0.8)\dagger}$ | $18.4^{(\pm 0.7)}$ |
| antsoccer arena stitch | $17.0^{(\pm 1.2)}$ | $24.5^{(\pm 2.8)}$ | $0.7^{(\pm 0.1)}$ | $2.1^{(\pm 0.1)}$ | $21.4^{(\pm 1.1)}$ | $0.8^{(\pm 0.2)}$ |
| humanoidmaze medium stitch | $46.1^{(\pm 1.9)}$ | $29.0^{(\pm 1.7)}$ | $36.2^{(\pm 0.9)}$ | $12.1^{(\pm 1.1)}$ | $12.3^{(\pm 0.6)}$ | $18.0^{(\pm 0.7)}$ |
| humanoidmaze large stitch | $8.6^{(\pm 1.4)}$ | $5.6^{(\pm 1.0)}$ | $4.0^{(\pm 0.2)}$ | $0.5^{(\pm 0.1)}$ | $1.2^{(\pm 0.2)}$ | $3.5^{(\pm 0.5)}$ |
| antmaze large navigate | $35.4^{(\pm 1.8)}$ | $24.0^{(\pm 0.6)}$ | $82.8^{(\pm 1.4)}$ | $34.2^{(\pm 1.3)}$ | $15.7^{(\pm 1.9)}$ | $74.6^{(\pm 2.3)}$ |
| cube single noisy | $9.2^{(\pm 0.9)}$ | $8.4^{(\pm 1.0)}$ | $38.3^{(\pm 0.6)}$ | $99.3^{(\pm 0.2)}$ | $70.6^{(\pm 3.3)}$ | $25.5^{(\pm 2.1)}$ |

RL methods with a separate value network to update the actor are in gray.
* The best non-RL methods up to significance are highlighted.     † We **bold** the best performance.

## 4.4 Testing Compositionality in Simulation

We also tested the compositional behavior in simulation using tasks from the OGBench [14] offline RL benchmark suite. This environment features environments for locomotion and manipulation, each with multiple offline datasets that can be used for training, including one that explicitly tests compositional generalization (the "stitch" datasets) by creating multiple short datasets that comprise a single, larger task. Some environments can be seen in Fig. 5 We modify TRA to account for the lack of language instructions. See Appendix G for details.

We evaluate the performance of TRA on seven different environments in OGBench. In 5 of these environments we use the "stitch" dataset, while two other environment use a more general goal-reaching dataset ("navigate" and "noisy"). Table 2 shows the performance of TRA compared to other non-hierarchical methods on these environments from OGBench. Consistent with our real-world results Table 1 and Fig. 4, TRA outperforms other imitation and offline RL methods on certain environments that require compositional generalizations, including CRL [78] that also has a separate value and critic network.

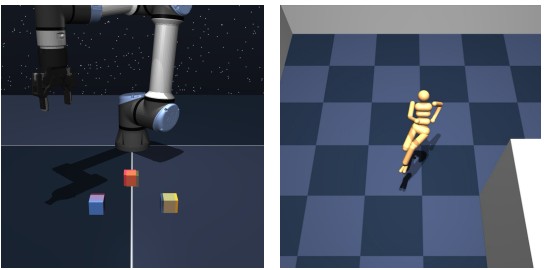

Figure 5: Two environments from the OGBench suite [14]. *Left:* a cube stacking environment. *Right:* a humanoid maze navigation environment.

In non-stitching environments, while traditional offline RL methods outperform TRA, TRA still improves on GCBC.

## 4.5 Failure Cases

Since we use a Gaussian policy, TRA struggles when multimodal behavior is observed, and sometimes fails to reach the goal due to early grasping or incorrect reaching [37]. While TRA did seem to provide small improvements on the in-distribution tasks of Ⓐ, the primary benefits derived from TRA were seen on compositionally-OOD tasks. We further discuss failure cases in Appendix E.1.

## 5 Conclusions and Limitations

In this paper, we studied a temporal alignment objective for the representations used in (goal- and language-conditioned) behavior cloning. This additional structure provides robust compositional generalization capabilities in both real-world robotics tasks and simulated RL benchmarks. Perhaps surprisingly, these results suggest that generalization properties usually attributed to reinforcement learning methods may be attainable with supervised learning with well-structured, temporally-consistent representations.

**Limitations and Future Work** While TRA consistently outperformed behavior cloning in real world and simulation evaluations, the degree of improvement degrades when behavior cloning cannot solve the task at all. Future work could examine how to improve compositional generalization in such cases through additional structural constraints on the representation space. To scale to more complex settings, similar approaches with more complex architectures such as transformers and diffusion policies may be needed for policy and/or representation learning. Methods like TRA that learn compositional task representations could be used with more complex models like VLMs [73], VLAs [79], or LLMs [80] to improve their generalization capabilities. In these settings, future work should examine safety of task generalization when interacting with humans and avoid emergent misalignment [81].

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

## A  Code and Website

An implementation of TRA is available at `https://anonymous.4open.science/r/ogcrl-43A4/`.

## B  TRA Implementation

In this section, we provide details on the implementation of temporal representation alignment (TRA) and its training process.

### B.1  Dataset Curation

We use an augmented version of BridgeData. We augment the dataset by rephrasing the language annotations, as described by [27], with 5 additional rephrased language instruction for each language instruction present in the dataset, and randomly sample them during training.

During data loading process, for each observation that is sampled with timestep $k$, we also sample $k^+ \triangleq \min(k + x, H), x \sim \text{Geom}(1 - \gamma)$, and load $s_k$ along with $s_{k^+}$. We employ random cropping, resizing, and hue changes during training process image robustness. We set $\gamma = 0.95$ for policy training on BridgeData.

### B.2  Policy Training

We use a ResNet-34 architecture for the policy network. We train our policy with one Google V4-8 TPU VM instance for 150,000 steps, which takes a total of 20 hours. We use a learning rate of $3 \times 10^{-4}$ with the ADAM optimizer [82], 2000 linear warm-up steps, and a MLP head of 3 layers of 256 dimensions after encoding the observation representations as well as goal representations.

## C  Baseline Implementations

We summarize the implementation details of the baselines discussed in Section 4.2.

### C.1  Octo

We use the Octo-base 1.5 model publicly available on HuggingFace for evaluating Octo baselines. We use inference code that is readily available for both image- and language- conditioned tasks. During inference, we use an action chunking window of 4 and an execution horizon window of 4.

### C.2  Behavior Cloning

We use the same architecture for LCBC and GCBC as in Walke et al. [13], Myers et al. [27]. During the training process we use the same hyperparameters as TRA.

### C.3  Advantage Weighted Regression

In order to train an AWR agent without separately implementing a reward critic, we follow Eysenbach et al. [78] and use a surrogate for advantage:

$$\mathcal{A}(s_t) = \mathcal{L}_{\text{NCE}}\big(f(s_t), f(g)\big) - \mathcal{L}_{\text{NCE}}\big(f(s_{t+1}), f(g)\big). \tag{18}$$

Here, $f$ can be any of the encoders $\phi$, $\xi$, $\psi$. $\mathcal{L}$ is the same InfoNCE loss defined Section 3, and $g$ is defined as either the goal observation or the goal language instruction, depending on the modality.

And we extract the policy using advantage weighted regression (AWR) [83]:

$$\pi \leftarrow \arg\max_{\pi} \mathbb{E}_{s,a\sim\mathcal{D}} \Big[\log \pi(a|s,z) \exp\big(A(s,a)/\beta\big)\Big]. \tag{19}$$

During training, we set $\beta$ to 1, and we use a batch size of 128, the same value as policy training for our method.

## D  Experiment Details

In this section, we go through our experiment details and how they are set up. During evaluation, we randomly reset the positions of each item within the table, and perform 5 to 10 trials on each task, depending on whether this task is important within each scene. We examine tasks that are seen in BridgeData, which include conventionally less challenging tasks such as object manipulation, and challenging tasks to learn within the dataset such as cloth folding and drawer opening.

### D.1  List of Tasks

Table 3 describes each task within each scene, and the language annotation used when the policy is used for inference. Every task that is outside of the drawer scene are multiple step, and require compositional generalization.

### D.2  Inference Details

During inference, we use a maximum of 200 timesteps to account for long-horizon behaviors, which remains the same for all policies. We determine a task as successful when the robot completes the task it was instructed to within the timeframe. For evaluating baselines, we use 5 trials for each of the tasks.

## E  Additional Visualizations

In this section, we show additional visualizations of TRA's execution on compositionally-OOD tasks. We use *folding, taking mushroom out of the drawer*, and *corn on plate, then sushi in the pot* as examples which require compositionality Fig. 6 and Appendix E.

Table 3: Task Instructions

| Scene | Count | Task Description | Instruction |
|---|---|---|---|
| Drawer | 10 | open the drawer | "open the drawer" |
| | 10 | put the mushroom in the drawer | "put the mushroom in the drawer" |
| | 10 | close the drawer | "close the drawer" |
| Task Generalization | 10 | put the spoons on the plates | "move the spoons onto the plates." |
| | 10 | put the spoons on the towels | "move the spoons on the towels" |
| | 10 | fold the cloth into the center from all corners | "fold the cloth into center" |
| | 10 | sweep the towels to the right | "sweep the towels to the right of the table" |
| Semantic Generalization | 10 | put the sushi and the corn on the plate | "put the food items on the plate" |
| | 10 | put the sushi and the mushroom in the bowl | "put the food items in the bowl" |
| | 10 | put the sushi, corn, and the banana in the bowl | "put everything in the bowl" |
| Tasks With Dependency | 10 | take mushroom out of drawer | "open the drawer and then take the mushroom out of the drawer" |
| | 10 | move bell pepper and sweep towel | "move the bell pepper to the bottom right corner of the table, and then sweep the towel to the top right corner of the table" |
| | 10 | put the corn on the plate, *and then* put the sushi in the pot | "put the corn on the plate and then put the sushi in the pot" |

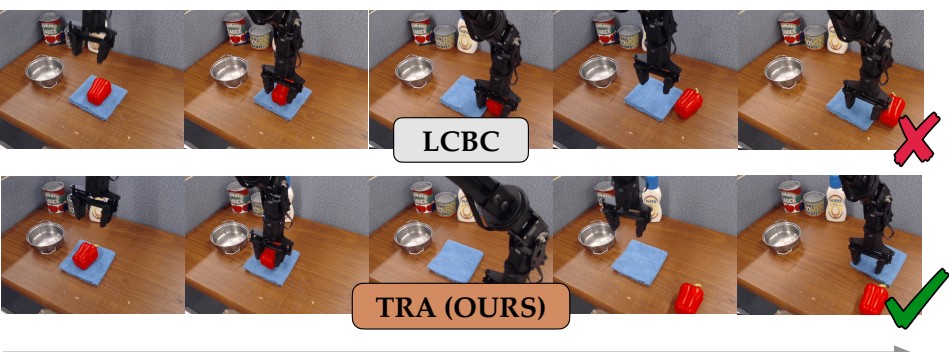

Figure 6: Example rollouts of a task with TRA and LCBC. While TRA is able to successfully compose the steps to complete the task, LCBC fails to ground the instruction correctly.

### E.1 Failure Cases

We break down failure cases in this section. While TRA performs well in compositional generalization, it cannot counteract against previous failures seen with behavior cloning with a Gaussian Policy.

## F Analysis of Compositionality

We prove the results from Section 3.5.

### F.1 Goal Conditioned Analysis

**Theorem 1.** *Suppose $\mathcal{D}$ is distributed according to Eq. (12) and $\mathcal{D}^*$ is distributed according to Eq. (12). When $\gamma > 1 - 1/H$ and $\alpha > 1$, for optimal features $\phi$ and $\psi$ under Eq. (11), we have*

$$\text{ERR}(\pi; \mathcal{D}^*) \leq \text{ERR}(\pi; \mathcal{D}) + \frac{\alpha - 1}{2\alpha} + \left(\frac{\alpha - 2}{2\alpha}\right) \mathbb{1}\{\alpha > 2\}. \quad (17)$$

*Proof.* We have from Eq. (14) for $K \sim \text{Geom}(1 - \gamma)$:

$$\text{ERR}(\pi; \mathcal{D}^*) \triangleq \mathbb{E}_{\mathcal{D}^*} \left[ \frac{1}{H'} \sum_{t=1}^{H'} \frac{\|\tilde{a}_{t,i} - \pi(\tilde{s}_{t,i}, \tilde{g}_i)\|^2}{n_{d_\mathcal{A}}} \right]$$

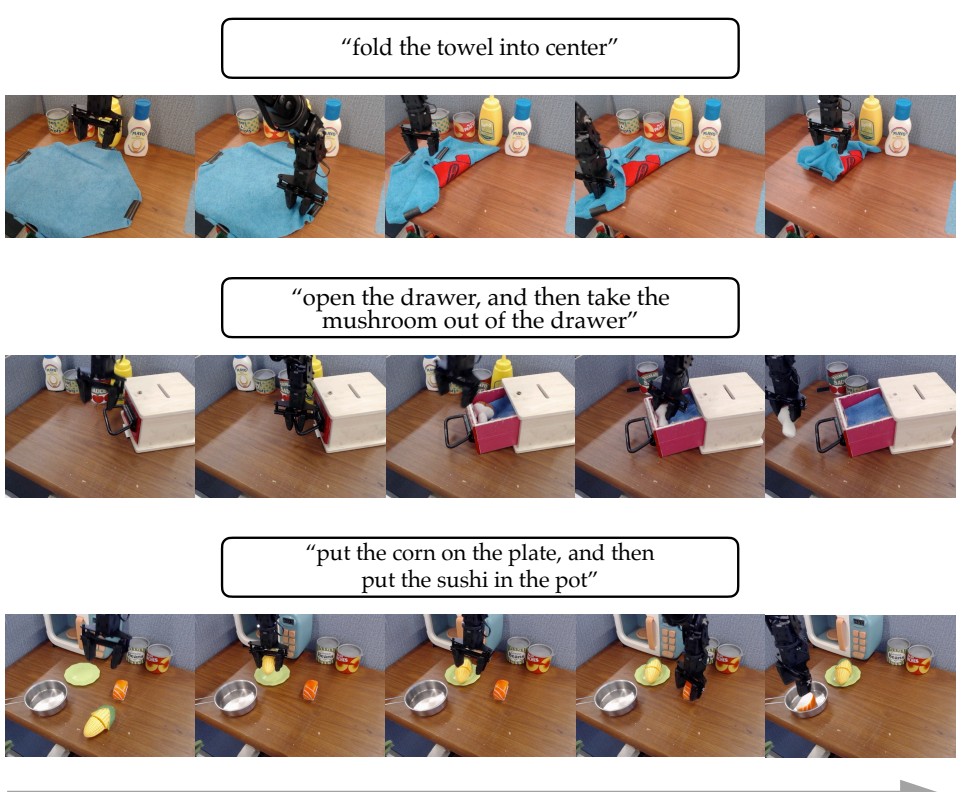

Figure 7: TRA performs compositional generatlization over a variety of tasks seen within BridgeData.

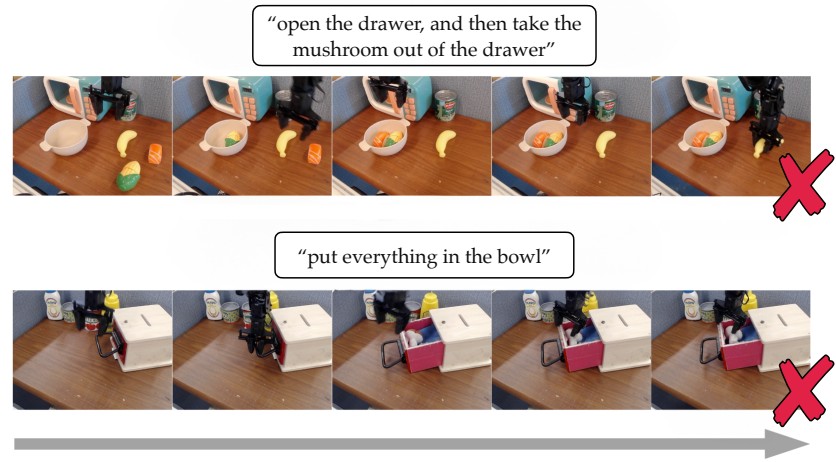

Figure 8: Most of the failure cases came from the fact that a policy cannot learn depth reasoning, causing early grasping or late release, and it has trouble reconciling with multimodal behavior.

$$= \frac{1}{H'} \mathbb{E}_{\mathcal{D}^*}\Big[\sum_{t=1}^{H'-2H} \frac{\|\tilde{a}_{t,i}-\pi(\tilde{s}_{t,i},\tilde{g}_i)\|^2}{n_{d_{\mathcal{A}}}}\Big] + \frac{1}{H'} \mathbb{E}_{\mathcal{D}^*}\Big[\sum_{H'-2H+1}^{H'-H} \frac{\|\tilde{a}_{t,i}-\pi(\tilde{s}_{t,i},\tilde{g}_i)\|^2}{n_{d_{\mathcal{A}}}}\Big]$$

$$+ \frac{1}{H'} \mathbb{E}_{\mathcal{D}^*}\Big[\sum_{t=H'-H+1}^{H'} \frac{\|\tilde{a}_{t,i}-\pi(\tilde{s}_{t,i},\tilde{g}_i)\|^2}{n_{d_{\mathcal{A}}}}\Big]$$

$$\leq \frac{1}{H'} \mathbb{E}_{\mathcal{D}^*}\Big[\sum_{t=H'-H+1}^{H'} \frac{\|\tilde{a}_{t,i}-\pi(\tilde{s}_{t,i},\tilde{g}_i)\|^2}{n_{d_{\mathcal{A}}}}\Big] + \frac{1}{H'} \mathbb{E}_{\mathcal{D}^*}\Big[\sum_{t=H'-2H+1}^{H'-H} \frac{\|\tilde{a}_{t,i}-\pi(\tilde{s}_{t,i},\tilde{g}_i)\|^2}{n_{d_{\mathcal{A}}}}\Big]$$

$$+ \Big(\frac{\alpha-2}{2\alpha}\Big) \mathbb{1}\{\alpha > 2\}$$

$$\leq \frac{1}{H'} \mathbb{E}_{\mathcal{D}^*}\Big[\sum_{t=H'-H+1}^{H'} \frac{\|\tilde{a}_{t,i}-\pi(\tilde{s}_{t,i},\tilde{s}_{H',i})\|^2}{n_{d_{\mathcal{A}}}}\Big]$$

$$+ \frac{1}{H'} \mathbb{E}_{\mathcal{D}^*}\Big[\sum_{t=H'-2H+1}^{H'-H} \mathbb{E}_K\Big[\frac{\|\tilde{a}_{t,i}-p^\pi(\tilde{s}_{t,i}|\tilde{s}_{H'-K,i})\|^2}{n_{d_{\mathcal{A}}}}\Big]\Big] + \Big(\frac{\alpha-2}{2\alpha}\Big) \mathbb{1}\{\alpha > 2\}$$

$$\leq \frac{1}{H'} \mathbb{E}_{\mathcal{D}^*}\Big[\sum_{t=H'-H+1}^{H'} \frac{\|\tilde{a}_{t,i}-\pi(\tilde{s}_{t,i},\tilde{s}_{H',i})\|^2}{n_{d_{\mathcal{A}}}}\Big]$$

$$+ \frac{1}{H'} \mathbb{E}_{\mathcal{D}^*}\Big[\sum_{t=H'-2H+1}^{H'-H} \mathbb{E}_K\Big[\frac{\|\tilde{a}_{t,i}-p^\pi(\tilde{s}_{t,i}|\tilde{s}_{H'-K,i})\|^2}{n_{d_{\mathcal{A}}}}\Big]\Big] + \Big(\frac{\alpha-2}{2\alpha}\Big) \mathbb{1}\{\alpha > 2\}$$

$$\leq \frac{1}{H'} \mathbb{E}_{\mathcal{D}^*}\Big[\sum_{t=H'-H+1}^{H'} \frac{\|\tilde{a}_{t,i}-\pi(\tilde{s}_{t,i},\tilde{s}_{H',i})\|^2}{n_{d_{\mathcal{A}}}}\Big]$$

$$+ \frac{1}{H'} \mathbb{E}_{\mathcal{D}^*}\Big[\sum_{t=H'-2H+1}^{H'-H} \mathbb{E}_K\Big[\frac{\|\tilde{a}_{t,i}-p^\pi(\tilde{s}_{t,i}|\psi(\tilde{s}_{H'-K,i}))\|^2}{n_{d_{\mathcal{A}}}}\Big]\Big] + \Big(\frac{\alpha-2}{2\alpha}\Big) \mathbb{1}\{\alpha > 2\}$$

$$\leq \text{ERR}(\pi;\mathcal{D}^*) + \frac{1}{H'} \mathbb{E}_{\mathcal{D}^*}\Big[\frac{1-\gamma^H}{1-\gamma}\Big] + \Big(\frac{\alpha-2}{2\alpha}\Big) \mathbb{1}\{\alpha > 2\}$$

$$\leq \text{ERR}(\pi;\mathcal{D}^*) + \frac{\alpha-1}{2\alpha} + \Big(\frac{\alpha-2}{2\alpha}\Big) \mathbb{1}\{\alpha > 2\}. \tag{20}$$

$\square$

### F.2 Language Conditioned Analysis

**Corollary 1.1.** *Under the same conditions as Theorem 1,*

$$\text{ERR}^\ell(\pi;\mathcal{D}^*) \leq \text{ERR}^\ell(\pi;\mathcal{D}) + \frac{\alpha-1}{2\alpha} + \Big(\frac{\alpha-2}{2\alpha}\Big) \mathbb{1}\{\alpha > 2\}.$$

The proof is similar to Appendix F.1, but over the predictions of $\xi$ instead of $\psi$.

### F.3 Visualizing the Bound

We compare the bound from Theorem 1 with the "worst-case" bound of $\text{ERR}(\pi;\mathcal{D}^*) - \text{ERR}(\pi;\mathcal{D})$ in Fig. 9. The bound from Theorem 1 is tighter than the worst-case bound, and it shows that the compositional generalization error decreases as $\alpha$ increases.

## G  OGBench Implementation Details

To implement TRA in OGBench, which does not have a corresponding language label for all goal-reaching tasks, we make the following revision to TRA to accommodate the lack of a language task.

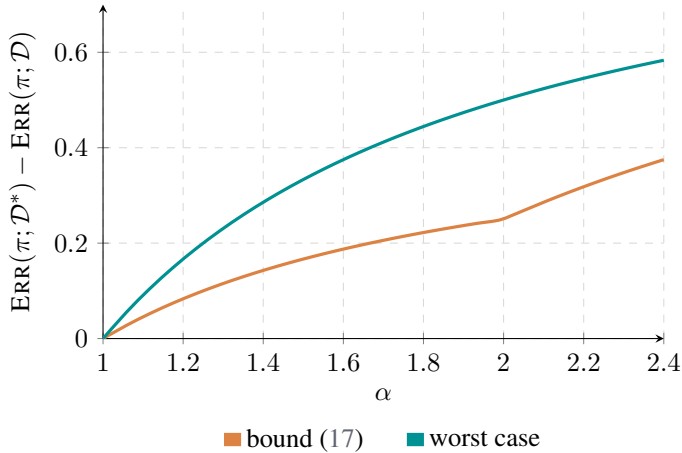

Figure 9: Visualizing the bound ([Eq. 17](#) from [Theorem 1](#)) on the compositional generalization error.

Table 4: **Success Rate for Different GCBC Architectures in OGBench.**

| Environment | GCBC | GCBC-$\phi$ |
|---|---|---|
| `antmaze medium stitch` | $\mathbf{45.5}^{\pm(\mathbf{3.9})}$ | $\mathbf{48.7}^{\pm(\mathbf{2.7})}$ |
| `antmaze large stitch` | $3.4^{\pm(1.0)}$ | $\mathbf{6.8}^{\pm(\mathbf{1.3})}$ |
| `antsoccer arena stitch` | $\mathbf{24.5}^{\pm(\mathbf{2.8})}$ | $1.4^{\pm(0.3)}$ |
| `humanoidmaze medium stitch` | $29.0^{\pm(1.7)}$ | $\mathbf{34.4}^{\pm(\mathbf{1.7})}$ |
| `humanoidmaze large stitch` | $\mathbf{5.6}^{\pm(\mathbf{1.0})}$ | $3.5^{\pm(1.1)}$ |
| `antmaze large navigate` | $\mathbf{24}^{\pm(\mathbf{0.6})}$ | $16.1^{\pm(0.8)}$ |
| `cube single noisy` | $8.4^{\pm(1.0)}$ | $\mathbf{8.7}^{\pm(\mathbf{0.9})}$ |

Table 5: **TRA hyperparameters.**

| Hyperparameter | Value |
|---|---|
| State and goal encoder dimensions | (64, 64, 64) |
| State and goal encoder latent dimension | 64 |
| Discount factor $\gamma$ | 0.995 (large locomotion environments), 0.99 (other) |
| Alignment coefficient $\alpha$ | 60 (medium locomotion), 100 (large locomotion), 20 (non-stitch) |

Table 6: Statistical comparisons of language-conditioned success rates

|  | Task | Best Score | TRA | GRIF | LCBC | Octo | AWR | TRA | GRIF | LCBC | Octo | AWR |
|---|---|---|---|---|---|---|---|---|---|---|---|---|
|  |  |  | t-values | | | | | p-values | | | | |
| **(A)** | open the drawer | 0.80 | **0.00** | 2.50 | **0.72** | **0.72** | **1.43** | **0.500** | 0.038 | **0.498** | **0.498** | **0.198** |
| **(A)** | mushroom in drawer | 0.80 | **0.00** | **0.00** | **1.43** | 6.00 | **0.72** | **0.500** | **1.000** | **0.198** | <0.001 | **0.498** |
| **(A)** | close drawer | 0.60 | **0.00** | **0.00** | **0.58** | **0.00** | **0.58** | **0.500** | **1.000** | **0.580** | **1.000** | **0.580** |
| **(D)** | take the item out of the drawer | 0.60 | **0.00** | 3.67 | 3.67 | **1.55** | 3.67 | **0.500** | 0.005 | 0.005 | **0.155** | 0.005 |
| **(B)** | put the spoons on towels | 0.80 | **0.00** | 1.43 | **2.50** | 6.00 | 2.50 | **0.500** | **0.198** | 0.038 | <0.001 | 0.038 |
| **(B)** | put the spoons on the plates | 0.90 | **0.00** | 3.13 | 3.13 | 3.13 | 9.00 | **0.500** | 0.020 | 0.020 | 0.020 | <0.001 |
| **(C)** | put the corn and sushi on plate | 0.90 | **0.00** | 9.00 | **1.89** | 9.00 | **2.06** | **0.500** | <0.001 | **0.113** | <0.001 | **0.058** |
| **(C)** | sushi and mushroom in bowl | 0.80 | **0.00** | 6.00 | **0.72** | 2.50 | **0.95** | **0.500** | <0.001 | **0.498** | 0.038 | **0.356** |
| **(C)** | corn, banana, and sushi in bowl | 0.80 | **0.00** | 6.00 | 6.00 | 6.00 | 3.18 | **0.500** | <0.001 | <0.001 | <0.001 | 0.005 |
| **(D)** | corn on plate then sushi in pot | 0.70 | **0.00** | 4.58 | **1.04** | **0.35** | **1.99** | **0.500** | 0.001 | **0.332** | **0.739** | **0.079** |
| **(A)** | sweep to the right | 0.80 | **0.00** | 2.50 | **1.43** | **1.43** | 6.00 | **0.500** | 0.038 | **0.198** | **0.198** | <0.001 |
| **(B)** | fold cloth into the center | 1.00 | **0.00** | 4.00 | **2.45** | **2.45** | **2.45** | **0.500** | 0.016 | **0.070** | **0.070** | **0.070** |
| **(B)** | move bell pepper and sweep towel | 0.50 | **0.00** | 3.00 | 3.00 | **1.15** | 3.00 | **0.500** | 0.015 | 0.015 | **0.277** | 0.015 |

For each task the reported t-values compare the candidate method to the best-performing method. We bold the t/p-values corresponding to methods that do not significantly ($p \geq 0.05$) differ from the best-performing method on each task.

We train a policy $\pi(a|\phi(s), \psi(g))$, in which we propagate the behavior cloning loss throughout the entire network. Both the state and goal encoders are MLPs with identical architecture. We detail the configuration in 5. This is to simulate the ResNet architecture and CLIP embeddings we use from real-world policy training. We define separate state and goal encoder $\phi(s)$ and $\psi(g)$, and we modify $\mathcal{L}_{\text{TRA}}$ as:

$$\mathcal{L}_{\text{TRA}} = \mathcal{L}_{\text{BC}}(\{s_i, a_i, s_i^+\}_{i=1}^K; \pi, \phi, \psi) + \alpha \mathcal{L}_{\text{NCE}}(\{s_i, s_i^+\}_{i=1}^K; \phi, \psi) \tag{21}$$

The rest of the implementation are carried over from OGBench. We evaluate each method with 10 seeds, and we take the final 3 evaluation epoch per seed to calculate the average success rate, the same way OGBench calculates success rate for its baselines. While we used $\alpha = 1$ in real world experiments, consistent with implementation from [27], we adjust our $\alpha$ value in OGBench, as it is a hyperparameter. We report our optimal $\alpha$ configuration in Table 5.

Note that $\alpha = 0$ turns the formulation into a version of GCBC with different architecture; we denote this GCBC-$\phi$. We compare the performance of GCBC and GCBC-$\phi$ here across the 7 environments using table 4. Although the second formulation is parameterized than the original GCBC configuration, they have similar performances across the environments that we have evaluated on — the performance of TRA does not rely on extra parameterization, but learning a structured temporal representation.

We report the value of hyperparameters in table 5. The rest of the relevant hyperparameters are implemented from OGBench unless specified in the table.

## H   Statistical Analysis

Statistical significance in Table 1 was computed with a one-sided Welch's t-test, comparing the success rates of the highest scoring method with the other methods for each task. Highlighting was then determined using a p-value of 0.05. The p- and t- values for the language conditioned evaluations in Table 1 are included in Table 6, and those for the goal conditioned evaluations in Table 7.

We also report the aggregated statistical comparisons of language-conditioned and image-conditioned performance across task sets in Table 8 and Table 9, respectively.

Table 7: Statistical comparisons of image-conditioned success rates

| | Task | Best Score | t-values | | | | | p-values | | | | |
|---|---|---|---|---|---|---|---|---|---|---|---|---|
| | | | TRA | GRIF | AWR | GCBC | Octo | TRA | GRIF | AWR | GCBC | Octo |
| (A) | open the drawer | 0.80 | **0.77** | **0.63** | **0.00** | **1.26** | **1.15** | **0.229** | **0.273** | **0.500** | **0.121** | **0.139** |
| (A) | mushroom in drawer | 0.90 | **0.00** | **1.89** | **1.13** | **0.45** | **0.00** | **0.500** | **0.057** | **0.152** | **0.335** | **0.500** |
| (A) | close drawer | 1.00 | **0.00** | 2.45 | **1.00** | 2.45 | 2.45 | **0.500** | 0.035 | **0.187** | 0.035 | 0.018 |
| (D) | take the item out of the drawer | 0.40 | **0.00** | 2.45 | 2.45 | 2.45 | **0.77** | **0.500** | 0.018 | 0.018 | 0.018 | **0.229** |
| (B) | put the spoons on towels | 1.00 | **0.00** | 4.00 | **1.63** | **1.63** | 3.67 | **0.500** | 0.008 | **0.089** | **0.089** | 0.003 |
| (B) | put the spoons on the plates | 0.90 | **0.00** | 9.00 | **0.45** | **1.89** | 9.00 | **0.500** | <0.001 | **0.335** | **0.057** | <0.001 |
| (C) | put the corn and sushi on plate | 0.70 | **0.00** | 4.58 | 1.85 | 1.99 | 4.58 | **0.500** | 0.001 | 0.040 | 0.040 | 0.001 |
| (C) | sushi and mushroom in bowl | 0.60 | **0.43** | 3.67 | **0.00** | 1.55 | **0.68** | **0.337** | 0.003 | **0.500** | 0.077 | **0.258** |
| (C) | corn, banana, and sushi in bowl | 0.50 | **0.00** | 3.00 | **0.00** | 3.00 | **0.34** | **0.500** | 0.007 | **0.500** | 0.007 | **0.372** |
| (D) | corn on plate then sushi in pot | 0.30 | **0.00** | 0.40 | 1.96 | 1.96 | 1.96 | **0.500** | 0.350 | 0.041 | 0.041 | 0.041 |
| (A) | sweep to the right | 0.80 | **0.40** | **1.26** | 4.00 | 4.00 | **0.00** | **0.350** | **0.121** | 0.008 | 0.008 | **0.500** |
| (B) | fold cloth into the center | 0.80 | **0.00** | 6.00 | 6.00 | 6.00 | **0.72** | **0.500** | <0.001 | <0.001 | <0.001 | **0.249** |
| (B) | move bell pepper and sweep towel | 0.60 | **0.00** | **1.55** | 3.67 | **1.55** | **0.68** | **0.500** | **0.077** | 0.003 | **0.077** | **0.258** |

For each task the reported t-values compare the candidate method to the best-performing method. We bold the t/p-values corresponding to methods that do not significantly ($p \geq 0.05$) differ from the best-performing method on each task.

Table 8: Statistical comparisons of language-conditioned performance aggregated across task sets

| Task Set | Best Score | t-values | | | | | p-values | | | | |
|---|---|---|---|---|---|---|---|---|---|---|---|
| | | TRA | LCBC | GRIF | Octo | AWR | TRA | LCBC | GRIF | Octo | AWR |
| (A) – One-Step | 0.77 | **0.00** | 2.38 | 2.38 | 2.78 | 3.22 | **0.500** | 0.0229 | 0.0229 | 0.0087 | 0.0028 |
| (B) – Task Concatenation | 0.80 | **0.00** | 5.36 | 5.36 | 5.36 | 6.25 | **0.500** | 0.0000 | 0.0000 | 0.0000 | 0.0000 |
| (C) – Semantic Generalization | 0.83 | **0.00** | 3.48 | -12.04 | 7.98 | 3.47 | **0.500** | 0.0021 | 0.0000 | 0.0000 | 0.0010 |
| (D) – Dependency | 0.65 | **0.00** | 2.61 | 5.94 | **1.27** | 3.71 | **0.500** | 0.0165 | 0.0000 | **0.2203** | 0.0010 |

For each set of tasks indicated, we report the t-values and p-values comparing the candidate method to the best-performing method, aggregated across all tasks in the set. We bold the t/p-values corresponding to methods that do not significantly ($p \geq 0.05$) differ from the best-performing method on each task set.

Table 9: Statistical comparisons of image-conditioned performance aggregated across task sets

| Task Set | Best Score | t-values | | | | | p-values | | | | |
|---|---|---|---|---|---|---|---|---|---|---|---|
| | | TRA | AWR | GCBC | GRIF | Octo | TRA | AWR | GCBC | GRIF | Octo |
| (A) – One-Step | 0.77 | **0.00** | 2.38 | 2.00 | 2.38 | **0.80** | **0.500** | 0.0115 | 0.0265 | 0.0115 | **0.2137** |
| (B) – Task Concatenation | 0.82 | **0.00** | 3.79 | 4.32 | 7.89 | 4.03 | **0.500** | 0.0003 | 0.0001 | 0.0000 | 0.0001 |
| (C) – Semantic Generalization | 0.57 | **0.00** | **0.77** | 3.35 | 6.16 | 2.00 | **0.500** | **0.2234** | 0.0009 | 0.0000 | 0.0270 |
| (D) – Dependency | 0.35 | **0.00** | 3.20 | 3.20 | **1.69** | **1.69** | **0.500** | 0.0024 | 0.0024 | **0.0519** | **0.0519** |

For each set of tasks indicated, we report the t-values and p-values comparing the candidate method to the best-performing method, aggregated across all tasks in the set. We bold the t/p-values corresponding to methods that do not significantly ($p \geq 0.05$) differ from the best-performing method on each task set.

