# OpenReview forum: "Temporal Representation Alignment: Successor Features Enable Emergent Compositionality in Robot Instruction Following"
_NeurIPS.cc/2025/Conference — NeurIPS 2025 poster_

### Official Review · Reviewer_Liiq · 2025-07-01

**Clarity:** 4
**Significance:** 3
**Originality:** 3
**Rating:** 5
**Confidence:** 3

**Summary:**

This paper proposes temporal representation alignment (TRA), a training approach for a language-conditioned and goal-conditioned policy which uses contrastive losses to align future (image-based) goal representations with current (image-based) state representations and text-based task specification representations. Intuitively, this enables the policy to be trained on two separate trajectories where Trajectory A’s goal state is the initial state of Trajectory B, then be able to compose these trajectories together conditioned only on Trajectory B’s goal state (or text instructions).

The paper draws from BridgeDataV2 to train TRA and several representative baseline policies for real-world execution. Evaluation task categories carefully separate tasks by their difficulty of compositional generalization, from simple one-step tasks to concatenated tasks, order-independent concatenated tasks with different objects of the same type (e.g., placing several food items into a container), and tasks with ordering dependencies between subtasks (e.g., opening a drawer then taking an object out of it). In these real-world experiments, the paper demonstrates that TRA achieves top success rates in all studied task categories in both language-conditioned and goal-conditioned settings. While policies from prior work are not statistically significantly worse than TRA in many cases, TRA achieves the best statistically significant performance in more categories than any other approach. Performance gains over the best baseline are most pronounced in tasks requiring compositional generalization. TRA vastly outperforms traditional offline reinforcement learning approaches (which are typically thought necessary to “stitch” subtasks together), and does not significantly benefit from being combined with such approaches. Together, these suggest that TRA is a remarkably effective approach despite its relative simplicity.

The paper goes on to evaluate TRA in 7 different simulated environments from OGBench. Consistent with the above results, TRA achieves the significantly best success rate in most environments (5 of 7), including 4 of 5 environments that require stitching together subtasks. The paper then briefly discusses TRA’s failure cases.

**Questions:**

Based on my Weakness 2, I ask one question to facilitate the rebuttal: **Do compositionally-OOD tasks truly consist of less common subtasks?** Given clarification on this in the rebuttal, it’s possible I will increase my quality score up to 4\.

My other weakness regarding Figure 1 cannot really be addressed without seeing a new version of the Figure, which the authors are welcome to discuss or present. That said, Figure 1 did not significantly impact my score (clarity score is already 4).

Given that the results are mixed in some cases (i.e., don’t consistently significantly outperform baselines), I don’t anticipate raising my significance score beyond 3 (which is a positive score). Given that the proposed approach is fairly simple, I don’t anticipate raising my originality score beyond 3 (which is a positive score). For these reasons, I don’t anticipate raising my overall score, which is already quite positive.

**Ethical Concerns:**

["NO or VERY MINOR ethics concerns only"]

**Final Justification:**

The concerns raised in my review have been addressed by the authors' response. However, given that the results are mixed in some cases (i.e., don’t consistently significantly outperform baselines), and given that the proposed approach is fairly simple, I will maintain my score as expected (although I do view it slightly more favorably now).

**Limitations:**

The discussion of limitations of the approach in Section 5 is sufficiently thoughtful and provides areas for future work to improve upon. The discussion of failure cases in the main body of the paper is quite brief. I don't see much discussion about the broader society impact in the paper.

**Paper Formatting Concerns:**

While the formatting generally looks acceptable, I provide a non-exhaustive list of additional minor writing suggestions and typos for the paper (which are not impacting my scores, and don’t require any response unless there are followup questions):

* Some figures, e.g., Figure 2-4, don’t seem to be referenced in the text describing related results (although most of them are positioned intuitively). For accessibility and easier reading, please clearly reference all figures and tables in the text.
* Some table captions could be more informative, e.g., Tables 1 and 2 should mention that the metric is success rate.
* Section 3 does not define some acronyms which may be unfamiliar to some readers: (GC)BC and NCE. Please define these in at least one place in the paper for broader readability.
* L17: Please explain what you mean by stitching here, as this term is reused throughout the paper and might not be clear to all readers.
* L23: I did not understand the phrase “which inference about which actions will lead to certain goals.” Did you intend to have a word like “enables” before “inference”?
* L24: The jump to successor representation learning is not clearly motivated by the first paragraph. I might recommend adding another sentence at the end of the first paragraph or beginning of second paragraph to motivate why we’re exploring this type of approach.
* L204: “detail” \-\> “detailed”
* L211: “1” \-\> “Table 1”?
* L242: “to to” \-\> “to”

**Quality:**

3

**Strengths And Weaknesses:**

**Strength 1: Paper is well written and clear.**

This paper is well written. It clearly conveys its motivation, contributions, methods, experimental design, results, and conclusions. The paper clearly and intuitively contextualizes its contributions with prior work in language- and goal-based robotic manipulation, compositional generalization in sequential decision making problems, and representation learning for physical states and procedures. It is quite polished with limited typos, and its figures and notations are mostly very clear and illustrative. Extensive appendices provide lower-level supplementary details about the datasets, methods, implementation, and analyses for reproducibility. In terms of clarity, it is among the top tier of submissions.

**Strength 2: Experiments soundly demonstrate effectiveness of TRA.**

The experimental results are motivated by clear and well-defined research questions. These questions are mostly effectively answered by experiments, demonstrating that TRA indeed improves zero-shot composition of tasks over prior methods, better captures rare skills in the dataset, and works even without explicit subtask planning or reinforcement learning. Performance improvements over prior work are sharp and significant. Experiments cover both language-conditioned and goal-conditioned execution, and simulated and physical settings. Evaluations carefully divide BridgeData according to difficulty in terms of compositional generalization. To the best of my knowledge, the baselines are sufficiently representative of approaches used in prior work.

**Weakness 1: Figure 1 does not provide a clear impression of the task or method.**

While this is a relatively minor concern, I found that Figure 1 left a lot of questions for me after reading Section 1 of the paper (where it’s referenced) for the first time. Specifically, I had the following questions (these do not require an answer, as I figured them out after reading the entire paper, but hopefully can help guide revisions of the figure):

* What is the task that TRA, AWR, and BC are trying to perform? The caption claims this is a “language task”, but it clearly involves manipulation too.
* What are we trying to observe through the comparison with AWR and BC? What are AWR and BC doing wrong, and how does it relate to compositionality? I see that AWR only puts one food in the bowl, but does this show the entire execution of the robot? Further, it seems that BC is unable to find any objects. What I don’t understand from the figure and caption is how this demonstrates a lack of “compositionality.”
* What is the TRA method doing? How is the time-contrastive loss calculated, and what does that have to do with the sequence of frames? Providing an intuitive sentence or phrase in the caption to describe the method at the high level would help a lot.

Also, the acronym for advantage weighted regression (AWR) is only defined in the appendix \- I would recommend defining this somewhere in the main body of the paper.

**Weakness 2: Paper seems to conflate “rarely-seen” and “challenging” tasks.**

One key question for the experimental results was “How well does TRA capture skills that are less common within the dataset?” However, the discussion around this at L241-246 seems to focus on the more challenging tasks in terms of compositional generalization (from Sets C and D involving semantic generalization across objects and subtask dependencies), which are compositionally OOD. However, can we safely assume that subtasks from Sets C and D are less common in training than others? I can’t seem to find details to support this, and would recommend that the authors better clarify this in the revised version of the paper, and consider rephrasing this research question if needed.

---

> ### Author Rebuttal · Authors · 2025-07-31
>
> Thank you for your detailed and constructive response. We will address some of your concerns below.
>
> > Weakness 2: Paper seems to conflate ''rarely-seen'' and ''challenging'' tasks ... Based on my Weakness 2, I ask one question to facilitate the rebuttal: Do compositionally-OOD tasks truly consist of less common subtasks? Given clarification on this in the rebuttal, it's possible I will increase my quality score up to 4.
>
> In practice, almost all rarely-seen tasks are challenging for policies trained on the Bridge v2 dataset [1]. The following table shows the frequency of instructions that are semantically similar to the evaluation tasks (same instruction label up to equivalent paraphrases) found in the dataset (of 53,191 instructions).
>
> | Scene | Command                                                                                              | Count | Percentage |
> |:-----:|:-----------------------------------------------------------------------------------------------------|------:|-----------:|
> | A     | open the drawer                                                                                      |   500 |    0.94% |
> | A     | take the mushroom out of the drawer                                                                  |     1 |    0.002% |
> | A     | close the drawer                                                                                     |   435 |    0.82% |
> | B     | put spoon on plate                                                                                   |    26 |    0.05% |
> | B     | put the spoon on the towel                                                                           |   155 |    0.291% |
> | B     | fold the cloth into the center                                                                       |    42 |    0.079% |
> | B     | sweep the towels to the right of the table                                                           |    72 |    0.135% |
> | C     | put the food items on the plate                                                                      |    41 |    0.077% |
> | C     | put the food items in the bowl                                                                       |    41 |    0.077% |
> | C     | put everything in the bowl                                                                           |     1 |    0.002% |
> | D     | open the drawer and then take the mushroom out of the drawer                                          |     0 |    0% |
> | D     | move the bell pepper to the bottom right corner of the table, and then sweep the towel to the top right corner of the table |     0 |    0% |
> | D     | put the corn on the plate and then put the sushi in the pot                                          |     0 |    0% |
>
> None of the tasks comprise more than 1% of the dataset. Note that while some demonstrations corresponding to tasks in scenes A, B, and C do appear in the Bridge data, the complex behaviors in scene D do not; they are **compositionally-OOD**. **Does this clarification address your concern?**
>
> > Weakness 1: Figure 1 does not provide a clear impression of the task or method. My other weakness regarding Figure 1 cannot really be addressed without seeing a new version of the Figure, which the authors are welcome to discuss or present. That said, Figure 1 did not significantly impact my score (clarity score is already 4).
>
> Thank you for your feedback. We will modify the caption to better define terms (language task, compositionality), and expand the TRA portion of the figure to show how frames are being aligned across time / with language. Please let us know if there are any other changes to the figure that might be helpful.
>
> > Missing references, typos, definitions in paper formatting concerns.
>
> Thank you for noticing these issues. We will correct them in our revision.
>
> ---
>
> [1] Walke, H. et al., 2023. BridgeData V2: A Dataset for Robot Learning at Scale. CoRL

---

> > ### Comment · Reviewer_Liiq · 2025-08-02
> >
> > Thanks for the response - your report of the data statistics indeed addresses my concern, and I may recommend adding this to your appendix and referencing it in a footnote. Your plan to expand Figure 1 sounds good to me.

---

> > > ### Author Response · Authors · 2025-08-07
> > > **Revision**
> > >
> > > Thank you for the suggestion—we will add this table and discussion to the appendix.

---

### Official Review · Reviewer_6ZRh · 2025-07-02

**Clarity:** 3
**Significance:** 2
**Originality:** 4
**Rating:** 4
**Confidence:** 3

**Summary:**

This paper addresses the problem of compositional generalization in robotic skills, where elementary skills learned during training can be recombined to solve longer, more complex tasks at test time. The authors propose Temporal Representation Alignment (TRA), a method that achieves compositionality purely through learning aligned state representations via temporal alignment of state and goal/instruction embeddings, which are then used in a goal-conditioned policy. To this end, a contrastive loss is introduced to align representations from the same expert demonstration. The policy is trained via behavioral cloning on expert demonstrations using these temporally aligned goal embeddings. TRA is evaluated in both simulation and real-world robotics settings, and the results show improved performance compared to various goal- or language-conditioned baselines, particularly in compositional generalization to novel task sequences.

**Questions:**

1. Compositionality-specific baselines: None of the baselines used are designed to explicitly address compositional generalization. Could the authors include comparisons to hierarchical methods or planning-based approaches, as discussed in the related work?
2. Clarification on rarely-seen skills: In the paragraph titled "Does TRA help rarely-seen skills within the dataset?", the actual frequency of the skills is not mentioned. How does the analysis support the claim about rarely-seen skills?
3. Task set clarifications: Are all tasks from set A in-distribution training tasks? What is the difference between task set B and C? Why is the use of different objects with the same type referred to as "semantic generalization"?

Minor Comments

- Table formatting errors: In Table 1, some rows highlight 0.00 success rates (e.g., row 4 for AWR, row 6 for Octo) as statistically significant best results. These seem to be mistakes, please double-check.
- Table references: Tables are referred to only by number in several places, which is confusing. For example, l. 211: "1 shows" → should be "Table 1 shows""; l. 611: "in 5" → should be "in Table 5".

**Ethical Concerns:**

["NO or VERY MINOR ethics concerns only"]

**Final Justification:**

The authors have appropriately addressed my questions regarding the dataset.
However, my main concern regarding baselines remains: while the paper compares against several strong methods, none are specifically designed to tackle compositional generalization. It would have been helpful to understand how TRA fares against such approaches.
That said, I find the paper to be a meaningful and promising contribution, and I believe its strengths outweigh these concerns.

**Limitations:**

yes

**Quality:**

3

**Strengths And Weaknesses:**

Strengths:
- addresses an important and challenging problem of compositional generalization in a novel way by introducing a temporal alignment loss over representations.
- empirical evaluation: includes both simulation and real-world experiments, and demonstrates statistically significant performance improvements over a variety of baselines (even though the baselines don’t directly target compositionality!).
- the paper is mostly well-written, and uses figures, algorithm boxes, and tables effectively to illustrate key concepts.

Weaknesses:
- some parts of the method description would benefit from more natural language explanation. For example, Eqs. 6 and 7 are presented without much intuition, and the roles of $\psi$ and $\phi$ (introduced in Eq. 5) are not clearly described in text until much later (Subsection 3.4).
- no baseline  is directly targeting compositional generalization. While the method is compared to strong goal- and language-conditioned policies, it is not evaluated against approaches that specifically tackle compositionality, such as hierarchical policies or planning-based methods, e.g. the methods discussed in the related work section.

---

> ### Author Rebuttal · Authors · 2025-07-31
>
> Thank you for your thoughtful feedback. We will address your questions and concerns in the section below. **Please let us know if you have any remaining questions or concerns.**
>
> > Clarification on rarely-seen skills: In the paragraph titled ''Does TRA help rarely-seen skills within the dataset?'', the actual frequency of the skills is not mentioned. How does the analysis support the claim about rarely-seen skills?
>
> The following table shows the frequency of the instruction labels in the Bridge dataset that are paraphrases of the commanded evaluation tasks (out of 53,191 instructions).
>
> | Scene | Command                                                                                              | Count | Percentage |
> | :---: | :--------------------------------------------------------------------------------------------------- | ----: | ---------: |
> | A     | open the drawer                                                                                      |   500 |     0.940% |
> | A     | take the mushroom out of the drawer                                                                  |     1 |     0.002% |
> | A     | close the drawer                                                                                     |   435 |    0.818% |
> | B     | put spoon on plate                                                                                   |    26 |    0.049% |
> | B     | put the spoon on the towel                                                                           |   155 |    0.291% |
> | B     | fold the cloth into the center                                                                       |    42 |    0.079% |
> | B     | sweep the towels to the right of the table                                                           |    72 |    0.135% |
> | C     | put the food items on the plate                                                                      |    41 |    0.077% |
> | C     | put the food items in the bowl                                                                       |    41 |    0.077% |
> | C     | put everything in the bowl                                                                           |     1 |    0.002% |
> | D     | open the drawer and then take the mushroom out of the drawer                                          |     0 |    0.000% |
> | D     | move the bell pepper to the bottom right corner of the table, and then sweep the towel to the top right corner of the table |     0 |    0.000% |
> | D     | put the corn on the plate and then put the sushi in the pot                                          |     0 |    0.000% |
>
> None of the tasks are more than 1% of the dataset, and for some tasks, only a few dozen related demonstrations exist out of almost 70,000 trajectories [1] (the discrepancy between the number of trajectories and instructions is from some of the dataset being unlabeled). While some similar behaviors to tasks in scenes A, B, and C appear in the dataset, none of the composed behaviors in scene D occur in the data.
>
> We will revise the paper to include this table and the discussion above.
>
> > Task set clarifications: Are all tasks from set A in-distribution training tasks? What is the difference between task set B and C? Why is the use of different objects with the same type referred to as ''semantic generalization''?
>
> For set A, both the drawer tasks are in the support of the dataset (see table above), though the actual evaluation scene features objects in different positions from those seen in the dataset. Only one trajectory that takes the mushroom out of the drawer is in the dataset.
>
> The main difference between set B and set C is that set B tasks involve performing the same subtask in sequence, but in set C, semantically distinct subtasks must be executed in sequence. For instance, the agent only needs to put multiple spoons on multiple towels in set B, but the agent needs to put multiple different kinds of food items in set C. Set C is more challenging than B because it depends on successful execution of distinct subtasks, and further requires subtasks to be executed in the correct order for success.
>
> > no baseline is directly targeting compositional generalization. While the method is compared to strong goal- and language-conditioned policies, it is not evaluated against approaches that specifically tackle compositionality, such as hierarchical policies or planning-based methods, e.g. the methods discussed in the related work section.
>
> We chose to focus this work on compositional generalization in non-hierarchical models, and thus excluded hierarchical and planning-based approaches. While often effective, these approaches introduce separate challenges, such as needing a much larger policy network [2], a small set of demonstrations during test-time [3], or access to powerful vision-language models [3,4]. Our approach highlights that compositionality can be achieved without using explicit planning or decomposition.
>
> > Table formatting errors
>
> > Table references: Tables are referred to only by number in several places ...
>
> Thank you for noting these errors, we have fixed them in the revision.
>
> ---
>
> [1] Walke, H. et al., 2023. BridgeData V2: A Dataset for Robot Learning at Scale. CoRL
>
> [2] Belkhale, S. et al., 2024. ''RT-H: Action Hierarchies Using Language''.
>
> [3] Myers, V. et al., 2024. ''Policy Adaptation via Language Optimization: Decomposing Tasks for Few-Shot Imitation''. CoRL.
>
> [4] Ahn, M. et al., 2022. ''Do As I Can, Not As I Say: Grounding Language in Robotic Affordances''. CoRL.

---

> > ### Comment · Reviewer_6ZRh · 2025-08-04
> > **Response to Rebuttal**
> >
> > Thank you for the detailed response and the clarifications regarding the task sets and instruction frequencies. I also appreciate the acknowledgment of related work on compositionality and the discussion of their limitations. I agree that these limitations make a case for using TRA over some of the more resource-intensive methods.
> >
> > However, I remain somewhat concerned that no baselines explicitly targeting compositional generalization, such as hierarchical or planning-based approaches, are included in the evaluation. The focus on non-hierarchical methods and exclusion of these baselines feels somewhat arbitrary. Including at least one comparison to such a baseline, would better contextualize the strengths or tradeoffs of TRA, and help clarify in which settings TRA offers advantages over other approaches. Even if TRA underperforms compared to some hierarchical or planning-based methods, that would be completely understandable given its advantages in simplicity and lower reliance on additional supervision or resources, it would still be valuable to see how it compares in practice.

---

> ### Author Response · Authors · 2025-08-07
> **Additional Revision**
>
> Thank you for continuing the discussion.
>
> For the goal-conditioned setting, we will add comparisons against HIQL [1], an explicitly hierarchical GCRL method that combines a high- and low-level policy. Reference results for HIQL are provided in the OGBench paper [2], which we will add to Table 2.
>
> For instruction following, numerous hierarchical/planning approaches have been proposed in recent years, exploiting the zero-shot capabilities of VLM/LLMs for task decomposition [3,4,5,6,7]. A simple hierarchical approach that has been applied to the Bridge [8] setting is to use a VLM to generate plans composed of low-level language subtasks (see [9], §4.3 under ''zero-shot''). We will add comparisons against this baseline on our real-world evaluation tasks in Table 1.
>
> We will also discuss the limitations of these hierarchical methods in our revised manuscript—namely, the need for extra parameters, increased latency at evaluation, and additional required domain knowledge (subgoal structure or access to powerful VLMs that can reason about plans).
>
> **Do these changes address this concern?**
>
> ---
>
> [1] Park, S. et al., 2023. ''HIQL: Offline Goal-Conditioned RL With Latent States as Actions.'' *NIPS*
>
> [2] Park, S. et al., 2025. ''OGBench: Benchmarking Offline Goal-Conditioned RL.'' *ICLR*
>
> [3] Ahn, M. et al., 2022. ''Do as I Can, Not as I Say: Grounding Language in Robotic Affordances.'' *CoRL*
>
> [4] Mees, O. et al., 2022. ''What Matters in Language Conditioned Robotic Imitation Learning Over Unstructured Data.'' *RAL*
>
> [5] Belkhale, S. et al., 2024. ''RT-H: Action Hierarchies Using Language.'' arXiv:2403.01823
>
> [6] Attarian, M. et al., 2022. ''See, Plan, Predict: Language-Guided Cognitive Planning With Video Prediction.'' arXiv:2210.03825
>
> [7] Michał, Z. et al., 2024. ''Robotic Control via Embodied Chain-of-Thought Reasoning.'' arXiv:2407.08693
>
> [8] Walke, H. et al., 2023. ''BridgeData V2: A Dataset for Robot Learning at Scale.'' *CoRL*
>
> [9] Myers, V. et al., 2024. ''Policy Adaptation via Language Optimization: Decomposing Tasks for Few-Shot Imitation.'' *CoRL*

---

### Official Review · Reviewer_3j6m · 2025-07-02

**Clarity:** 4
**Significance:** 3
**Originality:** 3
**Rating:** 5
**Confidence:** 4

**Summary:**

This paper introduces Temporal Representation Alignment (TRA), a novel approach for learning useful representations for solving goal/instruction-directed robotics tasks. The fundamental problem addressed by this paper is how to allow agents learning from a dataset of demonstrations to compose solutions to in-distribution tasks to solve longer-horizon out-of-distribution tasks. For example, if the dataset contains solutions to the tasks $s \rightarrow w$ and $w \rightarrow g$, how should an agent learn a representation that enables it to compose these solutions to solve the task $s \rightarrow g$?

The proposed approach treats this purely as a representation-learning problem, and solves it by ensuring that the representations of states $s$, $w$, and $g$ are aligned according to the temporal structure of the demonstration trajectories. By mapping states, goals, and natural-language instructions into a shared embedding space that captures this temporal alignment, the agent can naturally “stitch together” demonstrated behaviours to perform composite tasks.

The authors demonstrate the performance of TRA on a diverse set of long-horizon robotics tasks with positive results, particularly in cases where solving tasks requires the sort of behaviour composition that TRA is well-suited to.

**Questions:**

I found the discussion around Equation 4 – of how the proposed approach allows an agent to compose information from demonstration trajectories $s \rightarrow \ldots \rightarrow w$ and $w \rightarrow \ldots \rightarrow g$ to capture the higher-level behaviour $s \rightarrow \ldots \rightarrow g$ – to be particularly useful. This kind of sequential composition is a key idea motivating many existing hierarchical reinforcement learning methods. For instance, HIRO [1] learns a high-level policy that proposes a sequence of subgoals that a low-level policy navigates between, effectively stitching shorter segments together to solve a higher-level task.

**How does the proposed approach compare with hierarchical methods (like HIRO) that attempt to address this problem of sequentially composing (or, “stitching together”) behaviours?**

Please note that I raise this not to diminish this paper’s contributions; on the contrary, I believe such discussion would further highlight the originality and significance of the proposed approach and better set it apart from related work in hierarchical reinforcement learning.

---

While the authors target robotics problems with continuous state and action spaces, I suspect many of the ideas proposed in this paper might be more generally applicable.

**Do the authors foresee any challenges when adapting the proposed approach to work in domains with discrete state or action spaces?**
**If so, do they have any initial ideas about how these issues might be addressed?**

---

The reported success rates compare models trained on identical amounts of data, but it would be useful to know how the learned representations change as the amount of training data increases. If the proposed approach can learn useful representations with less (or more, as the case may be) data than alternative approaches, that would be useful to know.

**Have the authors investigated the sample-efficiency of the proposed method?**

---

**References**

[1] Nachum, O., Gu, S.S., Lee, H. and Levine, S., 2018. Data-efficient hierarchical reinforcement learning. Advances in neural information processing systems, 31.

**Ethical Concerns:**

["NO or VERY MINOR ethics concerns only"]

**Final Justification:**

Following discussion with the authors, I am happy to maintain my current rating for this paper and advocate for its acceptance.

I already had a favourable opinion of this paper: the proposed approach represents a step forward in representation learning for goal-directed tasks, and the evidence presented for its effectiveness is compelling.

During the discussion, the authors clarified additional properties of their method (e.g., natural extensibility to domains with discrete action spaces and favourable sample complexity), discussed additional experiments and comparisons (e.g., to hierarchical baselines), and addressed minor formatting and notational issues.

Like other reviewers, I initially questioned why no comparisons to hierarchical methods were included in the paper; I am happy to hear that the authors are going to include such comparisons in the camera-ready version. However, unlike some other reviewers, I believe the contributions and supporting evidence are already strong enough to merit acceptance without these additional results, which I view as a welcome addition rather than a hard requirement.

I also wish to note that some other reviewers commented on the proposed method not always significantly outperforming existing approaches across all evaluation domains. I do not support this criticism: it is not healthy for our community to dismiss novel ideas just because they are not dominant everywhere. In fact, I commend the rigour with which the authors performed their experiments and the candour with which they presented their results and limitations.

I thank the authors for their high-quality submission and their engagement during the rebuttal and discussion period.

**Limitations:**

The authors are candid about the limitations of the proposed approach, including a thoughtful discussion of these limitations and highlighting promising avenues for future work.

**Paper Formatting Concerns:**

There were no major formatting issues. However, I did notice a few minor issues:

- Line 110: The symbols $\phi, \psi,$ and $\xi$ are used before they are defined.

- Line 202: There’s a missing “a” or “the” in “…as a surrogate for value function.”

- Line 204: “A more detail approach…” $\rightarrow$ “A more detail**ed** approach…”

- Line 211: “  1 shows…” $\rightarrow$ “**Table** 1 shows…”


---

**Equation 1**

Although the authors’ intention is clear enough for me to understand the paper, I also had some minor notation comments and clarification questions about Equation 1.

On the first line, should $p(s_1 \mid s_1, a_1)$ instead be $\mathrm{P}(s_1 \mid s_1, a_1)$, the one-step transition dynamics of the MDP? If not, could the authors please clarify what $p$ denotes?

On the second line, the double integral is written as $\int_{A} \int_{S} p\left(s_{k+1} \mid s, a\right) \\, dp^\pi_k\left(s \mid s_1, a_1\right)  \\, d \pi \left(a \mid s\right)$ without explicitly including the measure terms for the continuous state and action spaces. Am I right in interpreting this as shorthand for $\int_{S} \int_{A} \mathrm{P}\left(s_{k+1} \mid s, a\right) \\, p^\pi_k\left(s \mid s_1, a_1\right) \\, \pi\left(a \mid s\right) \\, da \\, ds$? If so, I would recommend expressing the integral in this form to improve clarity.

On the third line, I am a little confused by the term $p^\pi(s_k \mid s_1, a_1)$. Specifically,

- why is it conditioned on the initial state-action pair $s_1, a_1$, and
- why is the time-index subscript on $p^\pi$ missing?

I would appreciate it if the authors could clarify what they mean here. Apologies if I am overlooking something obvious.

**Quality:**

4

**Strengths And Weaknesses:**

**Originality & Significance**

This paper addresses the important problem of allowing robotic agents to compose behaviours from large demonstration datasets to solve complex, long-horizon tasks that exceed the temporal scope of any single trajectory in their training data. By tackling this “trajectory stitching” problem directly and treating it as part of the representation learning problem, the proposed approach enables performance improvements over existing approaches across a wide range of challenging long-horizon tasks.

The proposed approach is also positioned well for widespread adoption. It is conceptually simple, seems straightforward to implement, and requires no additional information beyond what is already included in common behaviour-cloning datasets. Furthermore, its applicability is broadened by the fact that it can condition behaviour based on either raw goals or natural-language instructions.

Overall, I believe the ideas presented in this paper represent a significant step forward for goal-directed learning from demonstrations – and perhaps representation learning for sequential decision problems more broadly – and should prove a valuable tool for the community.

---

**Quality**

The quality of this submission is very high.

The evaluation is both thorough and focused. Centred around four specific research questions, it demonstrates the performance of the proposed approach across a diverse set of challenging real-world tasks, and includes comparisons to a wide range of relevant baselines.

It was particularly refreshing to see the honesty with which the authors presented their results. For instance, Table 1 highlights the best-performing methods *up to statistical significance*, rather than simply highlighting the absolute highest performance (which, in many cases, would have made the proposed approach look even better), and Section 4.5 explicitly discusses the proposed approach’s failure modes.

Alongside the thorough empirical evaluation, the method is also well-principled and supported by convincing theoretical work. I appreciated the care the authors took to introduce, motivate, and explain each element of their representation-learning objective.

---

**Clarity**

For the most part, the paper is written very clearly.

The authors explained the intuition behind the proposed approach well, and made good use of examples to help motivate it. The paper contains enough detail for me to understand the proposed method well. Taken together with the implementation details included in the appendices, I am confident that the authors have provided enough information to allow an expert reader to reproduce their results.

There are, however, a couple of potential issues with mathematical notation, which I have discussed in detail in the “Paper Formatting” section.

---

> ### Author Rebuttal · Authors · 2025-07-31
>
> Thank you for your thoughtful response. We will address the questions you have raised below. **Please let us know if you have any additional questions or concerns.**
>
> > How does the proposed approach compare with hierarchical methods (like HIRO) that attempt to address this problem of sequentially composing (or, ''stitching together'') behaviours?
>
> Hierarchical methods such as HIRO [1] often propose subgoals as high-level actions, represented as states to reach or, to instruct a low-level policy to reach goals. While hierarchical methods can yield impressive results, they introduce ''moving parts'' that can fail, and often require additional assumptions such as access to strong pretrained models [2,3]
>
> > Do the authors foresee any challenges when adapting the proposed approach to work in domains with discrete state or action spaces? If so, do they have any initial ideas about how these issues might be addressed?
>
> The method can be extended naturally to discrete environments. For discrete action spaces, a cross entropy loss could be used instead of MSE in the actor loss. Such an approach would also enable the use of autoregressive transformer-based architectures for the actor, that could be more effective in environments with multimodal behaviors. Likewise, discrete state spaces could be handled with simpler MLP architectures or through architectures like transformers.
>
> > Have the authors investigated the sample-efficiency of the proposed method?
>
> In BridgeData, we have trained TRA using 150,000 steps of training with a batch size of 128. Given that we sample an additional future state using geometrically for each data point, in the end we use a similar amount of data as the GCBC/LCBC baselines, which train for 300,000 steps with the same batch size. Octo [4] on the other hand, uses the entirety of OXE [5], which is much more data-intensive than the rest of the approaches.
>
> On OGBench, we use the same data loading procedure as CRL, which means that it requires double the amount of data required for GCBC, and the same amount of data as CRL.
>
> ---
>
> [1] Nachum, O. et al., 2018. ''Data-Efficient Hierarchical Reinforcement Learning''. NIPS.
>
> [2] Belkhale, S. et al., 2024. ''RT-H: Action Hierarchies Using Language''.
>
> [3] Ahn, M. et al., 2022. ''Do As I Can, Not As I Say: Grounding Language in Robotic Affordances''. CoRL.
>
> [4] Octo Model Team, 2024, ''Octo: an Open-Source Generalist Robot Policy''. RSS.
>
> [5] Open-X Embodiment Collaboration, 2024, ''Open-X Embodiment: Robotic Learning Datasets and RT-X Models''. ICRA.

---

> > ### Comment · Reviewer_3j6m · 2025-08-02
> > **Response to Rebuttal**
> >
> > I thank the authors for their thoughtful rebuttal - it directly and comprehensively addresses each of my questions.
> >
> > I believe the method's natural extensibility to discrete state/action settings and appealing sample efficiency compared to existing methods further strengthen the paper's contributions. It would be helpful to add comments on both of these points to the camera-ready version. I believe this will be helpful for readers interested in implementing or extending the proposed method in the future.
> >
> > On comparisons to hierarchical methods, I see your argument and understand your intention to frame this work as demonstrating that behaviour composition can be achieved without hierarchical explicit structures or planning. That said, I think many readers will nonetheless be interested in how the proposed method performs in comparison to representative hierarchical baselines. If you can think of a reasonable and correct way to include such a comparison, I think it would further underscore the significance of your work. If it turns out that a non-hierarchical methods with fewer "moving parts" and weaker assumptions can match or exceed the performance of a hierarchical one, I think that would be an important result to share with the community.
> >
> > To be clear, I do not think that the absence of these results is grounds for rejection. Your justification for omitting them is understandable, and I believe the paper is ultimately strong enough without them.
> >
> > One small clarification: please could you confirm whether my understanding of Equation 1 is correct, or indicate what I may have missed in relation to my comments in the "Paper Formatting Concerns" section?
> >
> > Finally, I was pleased to see favourable comments from the other reviewers. I will maintain my rating and will advocate for this paper's acceptance.

---

> > > ### Author Response · Authors · 2025-08-07
> > > **Response and Further Revisions**
> > >
> > > Thank you for your detailed response and continued engagement.
> > >
> > > > extensibility to discrete state/action settings
> > >
> > > Thank you for this suggestion. We will describe these extensions to discrete settings in our revised manuscript.
> > >
> > > > hierarchical baselines
> > >
> > > For goal-conditioned hierarchical methods, we will add comparisons against HIQL [1], a GCRL method that combines a high- and low-level policy decomposed subgoals. Reference results for HIQL are provided in the OGBench paper [2], which we will add to Table 2.
> > >
> > > Numerous hierarchical/planning approaches for following language instructions have been proposed in recent years, exploiting the zero-shot capabilities of VLM/LLMs for task decomposition [3,4,5,6,7]. A simple hierarchical approach that has been applied to the Bridge [8] setting is to use a VLM to generate plans composed of low-level language subtasks (see [9], §4.3 under ''zero-shot''). We will add comparisons against this baseline on the real-world evaluation tasks in Table 1.
> > >
> > > > Notation
> > >
> > > We apologize for these notational ambiguities—see the clarifications below. Please let us know if there are any further notational issues.
> > >
> > > > Equation 1
> > >
> > > Yes, your understanding is correct. The fully general $$\\int F(a)\\, \\mathrm{d}(\\pi(a))$$ notation is used to describe a measure-theoretic integral with respect to the probability distribution over actions induced by $\\pi$. When there is a natural reference measure over the action space (e.g., Lebesgue measure), this can equivalently be expressed as $$\\int F(a)\\, \\pi(a) \\, \\mathrm{d} a$$ We are happy to modify the definition to this second form for clarity.
> > >
> > > > $p$ vs $\\mathrm{P}$ in the 1-step dynamics
> > >
> > > Good catch, we will correct this notation to use $\\mathrm{P}$ consistently for the 1-step dynamics.
> > >
> > > > $p\_{k+t}^\\pi\\left(s\_{k+t} \\mid s\_t, a\_t\\right) \\triangleq p^\\pi\\left(s\_{k+1} \\mid s\_1, a\_1\\right)$
> > >
> > > Yes, the RHS should be $p^\\pi\_{k}$ as well. The intuition for this expression is it is saying the $k$-step forward dynamics are time-invariant (which follows from having stationary dynamics and policies). When combined with the other two lines, this gives a well-defined inductive construction for $p^{\\pi}\_{{t}}$ (each RHS term has smaller coefficients than the LHS terms).
> > >
> > > **Please let us know if there are any further issues or clarifications and we will be happy to make additional revisions.**
> > >
> > > ---
> > >
> > > [1] Park, S. et al., 2023. ''HIQL: Offline Goal-Conditioned RL With Latent States as Actions.'' *NIPS*
> > >
> > > [2] Park, S. et al., 2025. ''OGBench: Benchmarking Offline Goal-Conditioned RL.'' *ICLR*
> > >
> > > [3] Ahn, M. et al., 2022. ''Do as I Can, Not as I Say: Grounding Language in Robotic Affordances.'' *CoRL*
> > >
> > > [4] Mees, O. et al., 2022. ''What Matters in Language Conditioned Robotic Imitation Learning Over Unstructured Data.'' *RAL*
> > >
> > > [5] Belkhale, S. et al., 2024. ''RT-H: Action Hierarchies Using Language.'' arXiv:2403.01823
> > >
> > > [6] Attarian, M. et al., 2022. ''See, Plan, Predict: Language-Guided Cognitive Planning With Video Prediction.'' arXiv:2210.03825
> > >
> > > [7] Michał, Z. et al., 2024. ''Robotic Control via Embodied Chain-of-Thought Reasoning.'' arXiv:2407.08693
> > >
> > > [8] Walke, H. et al., 2023. ''BridgeData V2: A Dataset for Robot Learning at Scale.'' *CoRL*
> > >
> > > [9] Myers, V. et al., 2024. ''Policy Adaptation via Language Optimization: Decomposing Tasks for Few-Shot Imitation.'' *CoRL*

---

### Official Review · Reviewer_nx61 · 2025-07-02

**Clarity:** 3
**Significance:** 4
**Originality:** 3
**Rating:** 5
**Confidence:** 4

**Summary:**

This paper introduces Temporal Representation Alignment (TRA), an auxiliary contrastive objective that aligns representations of current states, future goals, and language instructions in a shared embedding space. By adding symmetric NCE losses on state–future‐state pairs and goal–language pairs to a standard behavior‐cloning policy, TRA enables zero‐shot “stitching” of known subtasks into novel, long‐horizon manipulation sequences. The authors demonstrate over 40 % relative gains on compositionally‐out‐of‐distribution tasks in a real WidowX250 setup (5 Hz execution) and show consistent improvements across seven environments in the OGBench benchmark—outperforming both imitation‐only and offline‐RL baselines on stitching tasks . A theoretical bound (Theorem 1) further relates in‐distribution to compositional error under successor‐feature assumptions.

**Questions:**

1. How does varying the geometric sampling parameter γ affect compositional success?

2. Can you provide t-SNE or clustering metrics showing that subtasks and their compositions occupy distinct, composable regions in the learned latent space?

3. Have you evaluated TRA on a different manipulator (e.g., a Panda arm) or at higher control rates?

4. How does TRA handle paraphrased or out-of-distribution instructions?

**Ethical Concerns:**

["NO or VERY MINOR ethics concerns only"]

**Final Justification:**

Thanks for the author's rebuttal. I don't have additional questions. I will keep my score.

**Limitations:**

Please see the weaknesses.

**Quality:**

4

**Strengths And Weaknesses:**

Strengths:

1. Minimal, Orthogonal Extension: TRA simply augments a standard goal- and language-conditioned BC policy with two lightweight contrastive losses, requiring no additional planner or value network.

2. Strong Compositional Gains: Real-world experiments show a > 40 % boost on out-of-distribution “stitch” tasks, and OGBench results confirm superior performance on environments demanding composition.

3. Unified Language + Vision: The same auxiliary losses work for both goal-image and natural-language instructions, demonstrating modality-agnostic compositionality.

4. Theoretical Backing: Theorem 1 and Corollary 1.1 provide clear guarantees that temporal alignment reduces compositional generalization error under mild assumptions.

Weaknesses:

1. Limited Physical-Robot Diversity: All hardware tests use a single 5 Hz WidowX250 arm. It remains unclear whether TRA scales to faster control loops (e.g., 20 Hz Panda) or different kinematics.

2. Some lacking ablation: Key choices—geometric horizon γ in the future‐state sampling, temperature and weighting of the contrastive losses—lack ablation.

3. Failure Modes Under Multimodality: As noted, TRA struggles with inherently multimodal tasks (e.g., when multiple valid grasps exist), but no mitigation strategies are proposed .

---

> ### Author Rebuttal · Authors · 2025-07-31
>
> Thank you for your thoughtful response. We will answer your questions below. **Please let us know if you have any additional questions or concerns.**
>
> > How does varying the geometric sampling parameter γ affect compositional success?
>
> We have run an additional ablation of $\gamma$ in OGBench in the `humanoidmaze-medium-stitch` environment, fixing the alignment coefficient at 40.
>
> | $\\gamma$      |   Mean (%) | Std. Err (%) |
> | :-----------: | ---------: | -----------: |
> | 0.8  |       28.3 |         2.2 |
> | 0.9  |       35.5 |         2.4 |
> | 0.95  |       42.8 |         2.6 |
> | 0.99  |       46.1 |         1.9 |
> | 0.995  |       41.8 |         2.2 |
>
> Values of $\\gamma$ between $\\approx 0.95$ to $0.995$ seem to perform well, which is similar to CRL's sampling coefficient of 0.995 [1].
>
> > Can you provide t-SNE or clustering metrics showing that subtasks and their compositions occupy distinct, composable regions in the learned latent space?
>
> Thank you for this suggestion. We cannot include figures in this response, but we will add this t-SNE visualization to our revised manuscript.
>
> > Have you evaluated TRA on a different manipulator (e.g., a Panda arm) or at higher control rates?
>
> BridgeData only contains one embodiment for training, which is the WidowX setup we have in our experiment. We will revise our future work section to discuss additional embodiments and datasets, such as DROID [2] or OXE [3].
>
> > How does TRA handle paraphrased or out-of-distribution instructions?
>
> We have done additional experiments on the task ''put the sushi, corn, and the banana in the bowl'', in which we have tried both ''put everything in the bowl'' and ''put the sushi, corn, and the banana in the bowl'' as our instructions. ''Put everything in the bowl'' has a success rate of 8 out of 10 tries, and ''put the sushi, corn, and the banana in the bowl'' has a success rate of 6 out of 10 tries.
>
> In practice, using an effective pretrained language encoder, such as the CLIP model we use, is the component that ensures robustness to these small perturbations in the instruction format.
>
> ---
>
> [1] Park, S. et al., 2025, ''OGBench: Benchmarking Offline Goal-Conditioned RL''. ICLR.
>
> [2] Khazatsky, A. et al., 2024, ''DROID: A Large-Scale In-the-Wild Robot Manipulation Dataset''. RSS
>
> [3] Open-X Embodiment Collaboration, 2024, ''Open-X Embodiment: Robotic Learning Datasets and RT-X Models''. ICRA

---

> > ### Comment · Reviewer_nx61 · 2025-08-06
> >
> > Thanks for your rebuttal. I don't have additional questions. I will keep my score.

---

### Note · Authors · 2025-08-16

Dear AC & reviewers,


Thank you for the thoughtful feedback and discussion. We were glad to see that all the reviewers gave overall positive evaluations of the work. Concerns mentioned in reviews included clarifications and requests for additional ablations and data analysis. We believe our responses have addressed these primary remaining concerns.

---

### Decision · Program_Chairs · 2025-09-17

**Decision:**

Accept (poster)

**Comment:**

In this paper, the authors propose Temporal Representation Alignment (TRA), a contrastive objective for robot learning that aligns state, goal, and instruction representations within a shared embedding space. The method leverages symmetric NCE losses between current–future states and between goals–language to a behavior-cloning policy, aiming to enable “stitching” of demonstrated subtasks into novel long-horizon tasks. Experiments in the paper show relative improvements on out of distribution tasks in real-world robot manipulation and consistent gains on benchmarks such as OGBench; particularly shining in tasks that require compositional generalization.

Initial reviews were positive. Strengths included the simplicity of the extension (adding two lightweight losses to normal behavior cloning), gains on compositional generalization on both real robots and simulation (R-nx61, R-3j6m), being applicable to both goal images and natural language (R-nx61), and theoretical guarantees (R-nx61). Weaknesses raised were the limited hardware diversity (all results being on a single arm running at 5 Hz, hence questions on whether it can scale to higher frequencies), missing ablations for hyperparameters such as the geometric horizon and loss weights, and observed failure modes on multimodal tasks.

The rebuttal provided ablations on geometric horizon, showing stable performance for different values, added t-SNE visualizations of representation clustering, and reported paraphrase robustness tests. Authors clarified sample efficiency (comparable or better than baselines at equal training steps) and argued that TRA naturally extends to discrete action/state settings. They acknowledged hardware limitations due to the choice of datasets used, and committed to discussing generalization to other embodiments in a future version.
Reviewers continued to be positive across the board post-rebuttal: R-nx61 maintained an “accept” recommendation, and R-3j6m has advocated for acceptance after noting clarifications on hierarchical comparisons, discrete extensions, and efficiency.

Overall, TRA introduces a simple auxiliary objective that improves compositional generalization without requiring planning or hierarchical structure. The paper is supported by both theoretical analysis and experimental results across simulation as wel as real world. The reviewers were aligned in support after rebuttal. I do recommend acceptance, but also reaffirm that adding comparisons with hierarchical baselines such as HIRO / HIQL would be a really useful addition to the camera-ready version.